# KNOWTRACE: EXPLICIT KNOWLEDGE TRACING FOR STRUCTURED RETRIEVAL-AUGMENTED GENERATION

## ABSTRACT

Recent advances in retrieval-augmented generation (RAG) furnish large language models (LLMs) with iterative retrievals of relevant information to strengthen their capabilities in addressing complex multi-hop questions. However, these methods typically accumulate the retrieved natural language text into LLM prompts, imposing an increasing burden on the LLM to grasp the *underlying knowledge structure* for high-quality multi-step reasoning. Despite a few attempts to reduce this burden by restructuring all retrieved passages or even entire external corpora, these efforts are afflicted with significant restructuring overhead and potential knowledge loss. To tackle this challenge, we introduce a new structured paradigm (KNOWTRACE) from the perspective of *explicit knowledge tracing*, which treats LLM as an agent to progressively acquire desired knowledge triplets during iterative retrievals and ultimately trace out a specific knowledge graph conditioned on the input question. This paradigm clearly unveils the logical relationships behind the unstructured text and thus can *directly facilitate LLM's inference*. Notably, it also naturally inspires a reflective mechanism of *knowledge backtracing* to identify supportive evidence and filter out useless retrievals in the correct trajectories, thus offering an effective way to *stimulate LLM's self-taught finetuning*. Extensive experiments demonstrate the superiority of our paradigm over three standard multi-hop question answering benchmarks. Our code is available at `https://github.com/xxrep/SRAG`.

## 1 INTRODUCTION

Large language models (LLMs) (Brown et al., 2020; Chowdhery et al., 2023; Touvron et al., 2023; Dubey et al., 2024) encapsulate a wealth of human knowledge into their massive parameters and have shown impressive performance across a wide range of language tasks through the form of question answering. Despite their remarkable capabilities, LLMs continue to struggle with factual errors (Welleck et al., 2020; Mallen et al., 2023; Zhang et al., 2023) when the input question exceeds their knowledge boundaries. As a prominent solution for this problem, Retrieval-Augmented Generation (RAG) (Lewis et al., 2020) empowers LLMs to incorporate new knowledge through the retrieval of relevant information. One-time retrieval-then-read (Borgeaud et al., 2022; Izacard et al., 2023) usually suffices to fulfill the information needs of single-hop questions, while the complex *multi-hop question answering task* still remains challenging due to their demands for intensive knowledge and multi-step reasoning capabilities, thus drawing significant attention within the research community.

To address the complex multi-hop questions, a feasible strategy is extending RAG into a multi-round iterative process (Trivedi et al., 2023; Press et al., 2023; Shao et al., 2023; Yao et al., 2023; Yang et al., 2024; Jiang et al., 2024). This process alternates between two stages: (1) performing partial LLM reasoning to guide subsequent retrieval, and (2) utilizing the retrieved information to enhance further reasoning, continuing until the available information is sufficient to deduce the final answer. Benefiting from such workflow of interleaving retrievals with LLM reasoning, the iterative RAG can narrow semantic gaps between the input questions and their requisite knowledge (Shao et al., 2023).

However, *more is not always better*—while the above iterative RAG methods can periodically bring in new external passages, they also present a great challenge for the LLMs in handling such ever-growing text due to the complexity and diversity of natural language expressions. Most of existing methods simply accumulate all these passages into LLM prompts, thereby struggling to perceive the *underlying knowledge structure* (i.e., logical connections among informative entities) for high-

quality multi-step reasoning (Braine, 1978). Several recent works try to mitigate this issue with an auxiliary process of reorganizing all retrieved passages (Li & Du, 2023; Cheng et al., 2024) or even entire external corpora (Edge et al., 2024; Sarmah et al., 2024) into specific structures (e.g., graphs and hierarchies), yet these approaches come with two inherent drawbacks: (1) they typically necessitate extensive LLM invocations for the intricate restructuring operations such as information extraction and refinement (Sarmah et al., 2024), thus incurring *significant computational overhead*; (2) such restructuring is decoupled from the question-specific reasoning process, potentially leading to the *loss of question-relevant knowledge* due to the lack of explicit reasoning guidance. In light of these limitations, a critical concern arises: *is there an elegant way to seamlessly incorporate the information restructuring process into the iterative RAG for higher-quality multi-step reasoning?*

In this paper, we give an affirmative response by introducing KNOWTRACE, a fresh RAG paradigm that can *coherently trace out question-specific knowledge structures* to bolster multi-step reasoning. At a high level, we draw upon a profound insight from constructivist theory (Fosnot, 2013): *learning is never merely about accumulating information, but involves actively absorbing crucial knowledge to construct and expand one's cognitive schema*. Inspired by this principle, KNOWTRACE leverages the LLM as an active knowledge organizer (rather than a passive information receiver) to explicitly trace question-relevant knowledge triplets from retrieved passages and progressively form a concrete knowledge graph (KG) for structured retrieval-augmented generation. More specifically, instead of simply accumulating textual information, our paradigm can be described as a reasoning-guided KG expansion process. As shown in Fig. 1(c), the LLM alternates between: (1) *knowledge exploration* to determine a set of entities along with their respective relation guidance based on the current KG (initially from the input question) for next retrieval; (2) *knowledge completion* to fill in these entity-relation pairs based on the retrieved passages for enriching the current KG—until it grasps adequate knowledge to solve the question or reaches the predefined maximum number of expansion rounds.

Owing to the perspective of explicit knowledge tracing, KNOWTRACE is able to adaptively maintain a transparent KG throughout the multi-step reasoning process for each input question. Such evident structure endows the LLM with an intelligible context to facilitate its inference capability, while also providing a graphical explanation of the entire reasoning trajectory. In particular, for the positive trajectories that ultimately arrive at correct answers, the acquired KGs can naturally induce a post-hoc reflective mechanism to identify the evidence subgraphs via *knowledge backtracing* (Fig. 1(c)). Notably, this offers a simple yet effective way to distill higher-quality reasoning rationales from the correct trajectories, enabling us to further improve KNOWTRACE in a *self-taught* manner. Different from recent self-taught methods (Zelikman et al., 2022; Yuan et al., 2023; Hosseini et al., 2024) that indiscriminately finetune on the entire correct trajectories, our paradigm can selectively filter out the procedural impurities (i.e., irrelevant knowledge and useless retrievals) for better self-improvement. Overall, by showcasing its dual advantages in multi-step inference and self-improvement, this work highlights the great significance of structured knowledge tracing for retrieval-augmented generation.

Our contributions are summarized as follows:

- We introduce a structured RAG paradigm (KNOWTRACE), which to the best of our knowledge is the first work to seamlessly enhance multi-step reasoning through *explicit knowledge tracing*.
- Based on the acquired question-specific knowledge structures (i.e., graphs), we further propose a post-hoc reflective mechanism (*knowledge backtracing*) to distill high-quality rationales from the correct trajectories, which can be used to guide the self-improvement of KNOWTRACE.
- We conduct extensive experiments on three multi-hop question answering benchmarks. Under different configurations of LLMs and retrieval models, KNOWTRACE consistently outperforms current RAG methods across all the datasets. The backtracing-guided finetuning further boosts the performance. Moreover, we also explore prompting strategies of this structured paradigm.

## 2  RELATED WORK

**Multi-hop question answering (MHQA).** This task involves answering the complex questions that require extensive knowledge and multi-step reasoning capability to arrive at a comprehensive answer (Yang et al., 2018; Ho et al., 2020; Trivedi et al., 2022). Different from the traditional approaches (Perez et al., 2020; Qi et al., 2020; Deng et al., 2022), this paper focuses on how to enhance the capabilities of LLMs to reason about complex multi-hop questions, which is consistent with the

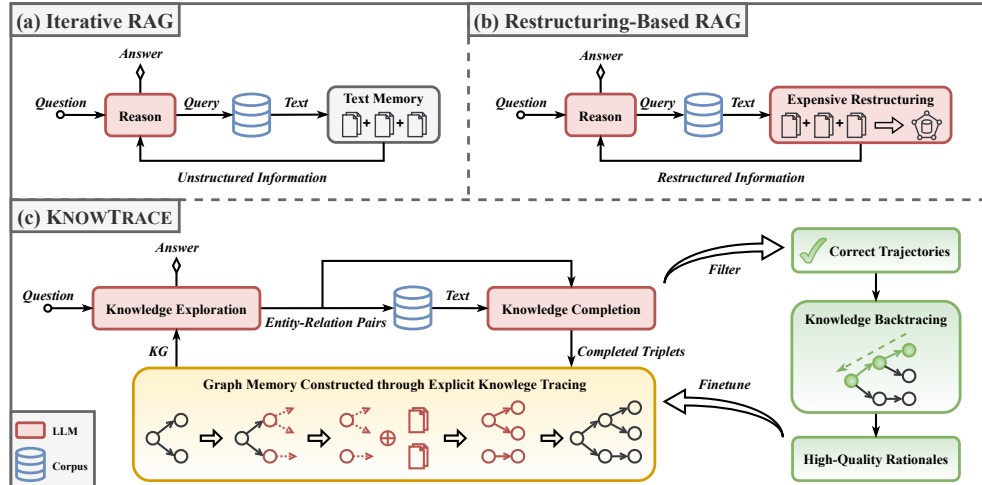

Figure 1: Overview of two representative workflows (a-b) and KNOWTRACE (c). Different from the unstructured RAG and restructuring-based RAG, KNOWTRACE progressively traces out a question-specific knowledge graph to facilitate multi-step reasoning, while also enabling a post-hoc reflective mechanism (knowledge backtracing) for self-taught finetuning.

recent retrieval-augmented generation methods (Trivedi et al., 2023; Press et al., 2023; Shao et al., 2023; Yao et al., 2023; Li & Du, 2023; Kim et al., 2024; Liu et al., 2024).

**Iterative RAG.** Retrieval-augmented generation (RAG) has been demonstrated as a promising technique to improve the performance of LLMs in knowledge-intensive NLP tasks (Lewis et al., 2020) (as shown in Fig. 1(a)). Early RAG approaches (Zhu et al., 2021; Lazaridou et al., 2022; Gao et al., 2023) typically conduct only one-time retrieval, which struggles to gather all information related to the input questions, especially for the complex multi-hop questions. To address this, a new genre of iterative RAG works has recently been developed. Self-Ask (Press et al., 2023) adopts a step-by-step approach to break down complex questions and solve each sub-question through the Google search. IRCoT (Trivedi et al., 2023) treats the output of each reasoning step as a retrieval query, collecting more passages to augment the subsequent reasoning steps. Similarly, Iter-RetGen (Shao et al., 2023) combines the reasoning output with the original question for further retrieval. Despite their prowess, these iterative methods inherently disregard the significance of the underlying knowledge structures behind the textual passages, which are essential elements of high-quality reasoning (Braine, 1978).

**Structure-enhanced RAG.** In light of the limitation of iterative RAG methods, several recent works employ an auxiliary restructuring process for retrieved passages (Li & Du, 2023; Panda et al., 2024; Liu et al., 2024; Cheng et al., 2024) or even entire text corpora (Edge et al., 2024; Sarmah et al., 2024; Peng et al., 2024). In particular, these methods first utilizes LLMs to conduct intricate restructuring operations such as entity recognition and relation extraction; according to the constructed knowledge structures, the LLMs then switches to inference mode for answering the input question. Unlike these methods that separate structuring from reasoning, our KNOWTRACE paradigm seamlessly integrates structuring and reasoning into a coherent process, thereby eliminating the limitations of substantial restructuring overhead and potential knowledge loss.

**Self-taught finetuning.** Self-taught finetuning provides an impressive way to enhance the reasoning capabilities of LLMs by finetuning them on their self-generated correct solutions (Zelikman et al., 2022; Yuan et al., 2023; Singh et al., 2024; Hosseini et al., 2024). Specifically, the self-improvement process (Zelikman et al., 2022; Hosseini et al., 2024) corresponds to a simple loop: use an LLM to infer a set of questions; collect all the generations that ultimately yield the correct answers; finetune on these collected data; restart to collect new generations from the newly finetuned LLM. Note that when a new dataset is collected, the model being finetuned is the original base LLM, rather than the one from the previous iteration. Such self-taught process is designed based on a priori assumption: the LLM generations that lead to correct answers reveal high-quality reasoning rationales. However, for the complex multi-step reasoning scenarios such as MHQA, a long-horizon reasoning trajectory may still contain irrelevant generations. In our paradigm, a reflective mechanism is naturally inspired to identify and filter out this noise for more effective self-improvement.

## 3 METHODOLOGY

### 3.1 OVERVIEW

This work introduces KNOWTRACE, a new structured paradigm that can self-organize the question-relevant knowledge structures (i.e., KGs) in a coherent manner to enhance the MHQA performance. As shown in Fig. 1, KNOWTRACE reformulates the LLM reasoner as an active knowledge organizer, which iteratively operates in two phases (*knowledge exploration* and *knowledge completion*) to trace the desired knowledge triplets, until an adequate KG is acquired to conclude with a definitive answer. We present a detailed description of this entire inference process in Section 3.2. Moreover, benefiting from the transparent graph structures, a post-hoc reflective mechanism (*knowledge backtracing*) is naturally inspired to filter out extraneous knowledge and futile retrievals from the correct trajectories. By leveraging the distilled high-quality rationales as procedural supervision, we can further improve KNOWTRACE via refined self-taught finetuning (Section 3.3). For the sake of clarification, all notations used in this paper are listed in Appendix A.

### 3.2 EXPLICIT KNOWLEDGE TRACING FOR STRUCTURED MULTI-STEP REASONING

Given a multi-hop question $q$ and a textual corpus $C$, KNOWTRACE autonomously acquires a set of $q$-relevant knowledge triplets from $C$ to form an explicit KG $\mathcal{G}_q = \{(e_s, r, e_o)|e_s, e_o \in \mathcal{E}_q, r \in \mathcal{R}_q\}$, in which $\mathcal{E}_q$ and $\mathcal{R}_q$ denotes the sets of entities and relations, respectively, and each triplet $(e_s, r, e_o)$ represents that there is a relation $r$ between subject entity $e_s$ and object entity $e_o$. The entire inference framework can be described as an iterative *explore-then-complete* process (Algo. 1), which consists of two LLM-driven phases: *knowledge exploration* and *knowledge completion*.

**Knowledge exploration.** During this phase, KNOWTRACE leverages the LLM's planning capability (Bohnet et al., 2024) to determine the actions for each iteration: either to provide a definitive answer as the final prediction or to explore more knowledge for KG expansion. Formally, at the $l$-th iteration ($1 \leq l \leq L$), KNOWTRACE integrates the input question $q$ and the KG $\mathcal{G}_q^{l-1}$ acquired from the previous $l-1$ iterations (initially $\mathcal{G}_q^0$ is empty) into an instruction prompt $I_{\text{exp}}$. This prompt is designed to elicit such a coherent generation from an LLM $M$: first, $M$ assesses whether $\mathcal{G}_q^{l-1}$ is adequate for deriving the final answer, and accordingly sets a boolean FLAG; if the FLAG is true, $M$ then outputs the answer $a$, along with a chain of thought $t$ (Wei et al., 2022) to reveal the final reasoning process; otherwise, $M$ then presents specific guidance on how to expand the current KG, that is, it adaptively determines the expansion points (i.e., entities) and the corresponding directions (i.e., relations), producing a set of entity-relation pairs that indicate the knowledge desired for the next reasoning step. We formulate this entire exploration process as follows:

$$\{\text{FLAG}, \mathcal{P}\} = M\left(I_{\text{exp}}\left(q, \mathcal{G}_q^{l-1}\right)\right), \tag{1}$$

where $\mathcal{P}$ is either the final prediction $[t, a]$ or the KG expansion guidance $\{(e_i, r_i)\}_{i=1}^P$ conditioned on the self-generated FLAG as described above. Note that $M$ can create new entities as the expansion points, rather than only selecting from the current KG. We refer to such entities as *initial entities*, as they typically correspond to the expansion beginnings of different components. After this phase, we then utilize each $(e_i, r_i)$ as the query to retrieve $N$ relevant passages from the textual corpus $C$, denoted as $\mathcal{C}_{(e_i, r_i)}^N$, while also employing this pair as a natural guidance for the subsequent phase.

**Knowledge completion.** Given the entity-relation pair $(e_i, r_i)$ along with retrieved passages $\mathcal{C}_{(e_i, r_i)}^N$, KNOWTRACE further harnesses the LLM's language understanding capability to purposefully grasp key knowledge from the unstructured text. Formally, with a completion instruction $I_{\text{com}}$ that receives $(e_i, r_i)$ and $\mathcal{C}_{(e_i, r_i)}^N$, the LLM $M$ is prompted to generate $(e_i, r_i)$-conditioned knowledge triplets:

$$\mathcal{T}_{(e_i, r_i)} = M\left(I_{\text{com}}\left(e_i, r_i, \mathcal{C}_{(e_i, r_i)}^N\right)\right). \tag{2}$$

If the passages $\mathcal{C}_{(e_i, r_i)}^N$ do not contain relevant information to $(e_i, r_i)$, $M$ can return an empty string. Each $(e_i, r_i)$ may also correspond to multiple knowledge triplets, i.e., $|\mathcal{T}_{(e_i, r_i)}| > 1$, showcasing the potential relation mapping properties (Li et al., 2022) behind natural language text. After completing each $(e_i, r_i)$ pair, KNOWTRACE induces a set of new knowledge triplets $\mathcal{T} = \bigcup_{i=1}^P \mathcal{T}_{(e_i, r_i)}$, thereby offering a more enriched KG $\mathcal{G}_q^l = \mathcal{G}_q^{l-1} \cup \mathcal{T}$ for the next iteration.

---

**Algorithm 1** Inference Process of KNOWTRACE

---

**Require:** Base LLM $M$; Prompt templates $\{I_{\texttt{exp}}, I_{\texttt{com}}\}$; Large-scale corpus $C$
**Input:** Question $q$
**Output:** Self-organized KG $\mathcal{G}_q$; Final thought process $t$; Predicted answer $a$

1: $\mathcal{G}_q^0 \leftarrow \emptyset$
2: **for** $l$ from $1$ **to** $L$ **do**
3:    $\{\texttt{FLAG}, \mathcal{P}\} \leftarrow M\left(I_{\texttt{exp}}(q, \mathcal{G}_q^{l-1})\right)$                 ▷ Knowledge Exploration (1)
4:    **if** $\texttt{FLAG}$ **then**
5:       $\mathcal{P}$ includes the thought process $t$ and the final answer $a$     ▷ Chain-of-Thought
6:       **return** $\mathcal{G}_q^{l-1}, t, a$
7:    **else**
8:       $\mathcal{P}$ includes a set of entity-relation pairs $\{(e_i, r_i)\}_{i=1}^P$
9:       **for** $i$ from $1$ **to** $P$ **do**               ▷ Parallelizable Inner Loop
10:         $(e_i, r_i)$ serves as a query for retrieving $\mathcal{C}_{(e_i,r_i)}^N$ from $C$
11:         $\mathcal{T}_{(e_i,r_i)} \leftarrow M\left(I_{\texttt{com}}\left(e_i, r_i, \mathcal{C}_{(e_i,r_i)}^N\right)\right)$     ▷ Knowledge Completion (2)
12:    $\mathcal{G}_q^l \leftarrow \bigcup_{i=1}^P \mathcal{T}_{(e_i,r_i)} \cup \mathcal{G}_q^{l-1}$            ▷ KG Expansion

---

**Knowledge prompting strategy.** Since KNOWTRACE aims to enhance LLM's inference with self-organized KG, one essential consideration lies in how to integrate KG information into LLM prompt. On this matter, we investigate three strategies to describe $\mathcal{G}_q$ for the prompt $I_{\texttt{exp}}$ in Eq. (1):

- *KG-to-Triplets.* We directly feed the elementary knowledge triplets into the LLM.

- *KG-to-Paths.* We combine the triplets that share common subject/object entities to form paths, and then regard these connected paths as the descriptions of KG.

- *KG-to-Text.* An additional generative model is employed to transform the acquired triplets into natural language, allowing the LLM to process the KG as standard text.

For our KNOWTRACE paradigm, we show that the *KG-to-Triplets* strategy offers the dual advantages of simplicity and effectiveness (Section 4.4).

**Connections to current RAG methods.** As illustrated in Fig. 1, the existing iterative RAG methods are either constrained by ever-increasing unstructured text or rely on costly restructuring operations, while the proposed KNOWTRACE moves beyond these methods by coherently tracing out a question-specific KG in the course of multi-step reasoning. This not only enables a clear knowledge structure to facilitate LLM's inference without the need for intricate restructuring operations, but also presents a succinct graph explanation of the entire reasoning process. Moreover, beyond these innate benefits for inference, our paradigm also possesses unique advantages in the self-improvement process, and we elaborate on this highlight in the following section.

### 3.3 REFLECTIVE KNOWLEDGE BACKTRACING FOR SELF-TAUGHT FINETUNING

Self-taught finetuning is an attractive process, in which the LLM can post-refine its own performance without human intervention. In line with the concept of self-training (Nigam & Ghani, 2000), recent techniques (Zelikman et al., 2022; Singh et al., 2024; Hosseini et al., 2024) improve the reasoning capabilities of LLMs by finetuning them on their own generations that arrive at the correct answers. Nevertheless, despite its effectiveness for one-time generation tasks, this process is inherently flawed when applied to recent RAG systems in MHQA scenario: for a complex multi-hop question, even if the final prediction is correct, the corresponding long-horizon reasoning trajectory may still contain multiple unnecessary LLM generations, which could diminish the efficacy of subsequent finetuning.

To address this limitation, the key challenge lies in how to grasp the reasoning rationales and remove the procedural impurities in correct trajectories. Based on our perspective of explicit knowledge tracing, the structured KNOWTRACE can progressively acquire a KG for the input question as reasoning progresses. In this regard, the acquired KG directly forms procedural records of the entire multi-step reasoning process. Such transparent records then naturally inspire us to design a post-hoc reflective mechanism (*knowledge backtracing*) to identify the supportive generations from correct trajectories.

---

**Algorithm 2** Backtracing-Guided Self-Training Process of KNOWTRACE

---

**Require:** Labeled dataset of question-answer pairs $\mathcal{D} = \{(q_d, \hat{a}_d)\}_{d=1}^D$
**Input:** Base LLM $M$
**Output:** Finetuned LLM $M_Z$
 1: $M_0 \leftarrow M$
 2: **for** $z$ **from** 1 **to** $Z$ **do**
 3:     $\mathcal{D}_z \leftarrow \emptyset$
 4:     **for** $d$ **from** 1 **to** $D$ **do**
 5:         $\{\mathcal{G}_{q_d}, t_d, a_d\} \leftarrow \texttt{KnowTrace}(M_{z-1}, q_d)$         ▷ Inference process (Algo. 1)
 6:         **if** $a_d == \hat{a}_d$ **then**
 7:             Collect input-output pairs $\{(I_{\texttt{exp}}(\cdot), \mathcal{P})\}$ and $\{(I_{\texttt{com}}(\cdot), \mathcal{T}_*)\}$
 8:             $\mathcal{S}_{q_d} \leftarrow \texttt{Backtracing}(\mathcal{G}_{q_d}, [t_d, \hat{a}_d])$     ▷ Knowledge backtracing (Section 3.3)
 9:             $\mathcal{P}^+ \leftarrow \texttt{Filter}(\mathcal{P}, \mathcal{S}_{q_d})$            ▷ Filter out unavailing exploration
10:             $\mathcal{T}_*^+ \leftarrow \texttt{Filter}(\mathcal{T}_*, \mathcal{S}_{q_d})$           ▷ Filter out extraneous completion
11:             $\mathcal{D}_z \leftarrow \mathcal{D}_z \cup \{(I_{\texttt{exp}}(\cdot), \mathcal{P}^+)\} \cup \{(I_{\texttt{com}}(\cdot), \mathcal{T}_*^+)\}$
12:     $M_z \leftarrow \texttt{Train}(M, \mathcal{D}_z)$         ▷ Finetune the base model on distilled generations

---

Formally, given a positive inference sample $(q, \mathcal{G}_q, [t, a])$ of KNOWTRACE that yields correct answer (i.e., the prediction $a$ exactly matches the ground-truth answer $\hat{a}$), we aim to grasp the key reasoning rationales from all LLM generations in Eq. (1) and (2). Due to the structured nature of our paradigm, the rationales of each sample $(q, \mathcal{G}_q, [t, a])$ essentially correspond to a knowledge subgraph $\mathcal{S}_q \subseteq \mathcal{G}_q$ that exactly supports the final prediction. In light of this, a simple yet effective backtracing process can be adopted for rationalization: first, since the ground truth label $\hat{a}$ could verify the rationality of the final LLM generation (i.e., $[t, a]$) (Wei et al., 2022; Xi et al., 2024), we accordingly select the entities that appear in $[t, a]$ as the informative *target entities*; then, we trace back along the graph structure of $\mathcal{G}_q$ from these entities to the *initial entities* (defined in Section 3.2), thereby inducing the expected subgraph $\mathcal{S}_q$ that consists of all the supportive knowledge. This subgraph showcases which triplets contribute to the final prediction, and thus offers guidance for filtering irrelevant generations:

○ (*Unavailing Exploration*) For each prompt-generation pair $(I_{\texttt{exp}}(\cdot), \mathcal{P})$ in Eq. (1), we remove $(e_i, r_i) \in \mathcal{P}$ (or even entire $\mathcal{P}$) that fails to produce any supportive knowledge triplets in $\mathcal{S}_q$.

○ (*Extraneous Completion*) For each prompt-generation pair $(I_{\texttt{com}}(\cdot), \mathcal{T}_*)$ in Eq. (2), we remove the generated knowledge triplets (or even entire $\mathcal{T}_*$) that are not included in $\mathcal{S}_q$.

In this way, the designed mechanism endows KNOWTRACE with the ability to reflect on reasoning trajectories and distill high-quality rationales. We incorporate this mechanism into the standard self-improvement technique (Zelikman et al., 2022), forming a backtracing-guided self-training process. As shown in Algo. 2, KNOWTRACE can bootstrap its reasoning ability through the following loop: (1) collect inference samples that lead to correct answers from a labeled MHQA dataset; (2) conduct backtracing mechanism to filter out irrelevant generations; (3) finetune the base LLM on the distilled generations for the next round of inference. We highlight that the backtracing mechanism is naturally built on the structures acquired by our KNOWTRACE, thus showcasing another unique advantage of the perspective of explicit knowledge tracing. In other words, KNOWTRACE not only self-organizes clear structures to enhance inference, but also offers an effective way to stimulate self-improvement.

## 4 EXPERIMENTS

To comprehensively validate the effectiveness of our proposals, we conduct extensive experiments, which are outlined as follows:

○ First, we compare the basic KNOWTRACE to a series of RAG approaches in the MHQA task (using two mainstream LLMs as reasoning backbones), aiming to demonstrate the facilitative effect of structured knowledge tracing on multi-step inference.

○ Second, we leverage the knowledge backtracing mechanism to stimulate self-taught finetuning, resulting in a new version called KNOWTRACE*. We present the positive effect of such mechanism from both performance and statistical perspectives.

○ Last but not least, we analyze the effect of configuring different retrieval methods and knowledge prompting strategies on the inference performance of KNOWTRACE.

## 4.1 EXPERIMENTAL SETUP

### 4.1.1 DATASETS AND METRICS

**Datasets.** We evaluate KNOWTRACE on three standard MHQA datasets in the open-domain setting: HotpotQA (Yang et al., 2018), 2WikiMultihopQA (2Wiki) (Ho et al., 2020), and MuSiQue (Trivedi et al., 2022). For each dataset, we follow (Trivedi et al., 2023; Li & Du, 2023) to randomly sample 100 questions from the development set for hyperparameter tuning, and another 500 questions are randomly sampled as the test set. For the open-domain corpora, we use Wikipedia 2017 (Yang et al., 2018) for HotpotQA and Wikipedia 2018 (Karpukhin et al., 2020) for the other two datasets. More details about these datasets can be found in Appendix B.

**Metrics.** We employ the exact match (EM) and the F1 score as evaluation metrics. The EM accuracy is calculated as the proportion of correct answers in the test set, where a prediction is deemed correct if it exactly matches one of the ground truth answers. The F1 score evaluates the overlap between the tokens in the prediction and the answer. We apply normalization to both the predictions and the answers when computing these two metrics, following the implementation of (Yao et al., 2023).

### 4.1.2 BASELINES

We compare KNOWTRACE with a series of advanced RAG approaches, which can be classified into three categories: (1) *one-time* retrieval-augmented chain-of-thought (Wei et al., 2022), i.e., RA-CoT; (2) *unstructured iterative approaches*: IRCoT (Trivedi et al., 2023), ReAct (Yao et al., 2023), Self-Ask (Press et al., 2023) and Iter-RetGen (Shao et al., 2023); (3) *restructuring-based approaches*: SG-Prompt (Li & Du, 2023) and ERA-CoT (Liu et al., 2024).

Here, we briefly describe two representative baselines related to our work. As an unstructured RAG approach, IRCoT (Trivedi et al., 2023) interleaves retrieval-augmented chain-of-thought reasoning and reasoning-guided retrieval until the final answer is reported or the maximum allowed number of reasoning steps is reached. A recent restructuring-based work, ERA-CoT (Liu et al., 2024), uncover the knowledge structure behind textual passages using a fully LLM-driven process: first, it identifies all entities involved in the text; then, it extracts both explicit and implicit relations between entities; next, it scores the reliability of the relations and removes those falling below a predefined threshold; after completing this intricate process, it performs the final answer prediction. More descriptions of all baselines can be found in Appendix C.

### 4.1.3 IMPLEMENTATION DETAILS

**Backbones.** We utilize `LLaMA3-8B-Instruct` (Dubey et al., 2024) as the base LLM $M$ for the main experiments, and also employ OpenAI's `gpt-3.5-turbo-instruct` (OpenAI, 2022) to investigate the effect of different LLM backbones. The design of LLM prompts (i.e., $I_{\mathrm{exp}}$ and $I_{\mathrm{com}}$) are presented in Appendix D for reproducibility. For each prompt, we provide four simple examples shared across all datasets to elicit the LLMs' instruction-following capability (Brown et al., 2020). We set the temperature of $0.0$ when calling the OpenAI's API, and use greedy decoding for LLaMA, to remove the effect of random sampling (Renze & Guven, 2024).

**Retrievers.** Under the open-domain setting, we employ entire Wikipedia dumps as retrieval corpora, and investigate three different retrieval methods to verify the compatibility of our proposal, including BM25 (Robertson et al., 2009), DPR (Karpukhin et al., 2020), and Contriever (Izacard et al., 2022). We perform BM25 retrieval with Elasticsearch (Gormley & Tong, 2015), and leverage BEIR (Thakur et al., 2021) framework for DPR and Contriever. In the main experiments, we retrieve $N = 5$ most relevant passages for each query with BM25, and also vary $N$ to $\{10, 20, 30, 50\}$ for further analysis.

**Self-taught finetuning.** For each dataset, we randomly sample 5,000 question-answer pairs to form $\mathcal{D}$ in Algo. 2. During self-improvement process, we utilize the proposed backtracing mechanism to collect supportive generations and augment finetuning dataset. The detailed statistical characteristics can be found in Section 4.3. Building upon the base LLM $M$, we train two distinct LoRA adapters (Hu et al., 2022) to specialize the capabilities of knowledge exploration and knowledge completion,

Table 1: Evaluation results on three multi-hop question answering datasets. We adopt two advanced LLMs as the backbones for each method, and select $N = 5$ most relevant passages for each retrieval. The best results are in **bold**, and the second best results are underlined.

| Methods | LLaMA3-8B-Instruct | | | | | | gpt-3.5-turbo-instruct | | | | | |
| | HotpotQA | | 2Wiki | | MuSiQue | | HotpotQA | | 2Wiki | | MuSiQue | |
| | EM | F1 | EM | F1 | EM | F1 | EM | F1 | EM | F1 | EM | F1 |
|---|---|---|---|---|---|---|---|---|---|---|---|---|
| RA-CoT | .206 | .314 | .194 | .255 | .132 | .203 | .308 | .429 | .272 | .364 | .164 | .258 |
| ReAct | .270 | .382 | .232 | .324 | .204 | .308 | .354 | .471 | .336 | .483 | .232 | .370 |
| IRCoT | .324 | .425 | .286 | .372 | .240 | .332 | .442 | .565 | .374 | .519 | .278 | .385 |
| Self-Ask | .252 | .367 | .218 | .325 | .186 | .275 | .352 | .468 | .328 | .464 | .204 | .323 |
| Iter-RetGen | .304 | .393 | .264 | .347 | .228 | .317 | .426 | .542 | .368 | .495 | .246 | .372 |
| SG-Prompt | .328 | .411 | .306 | .369 | .236 | .342 | .448 | .583 | .430 | .537 | .254 | .369 |
| ERA-CoT | .344 | .435 | .294 | .365 | .242 | .346 | .460 | .592 | .432 | .543 | .268 | .376 |
| KNOWTRACE | **.386** | **.479** | **.342** | **.403** | **.280** | **.387** | **.516** | **.633** | **.476** | **.582** | **.304** | **.425** |

respectively. We tune the training epoch in $\{1, 2, 3\}$, batch size in $\{32, 64, 128\}$, and learning rate in $\{1e{-}5, 5e{-}5, 1e{-}4, 3e{-}4\}$. We iteratively run this backtracing-guided self-improvement process for KNOWTRACE until the performance saturates, and then report the best results.

## 4.2 INFERENCE PERFORMANCE COMPARISON

Table 1 summarized the main experimental results on three standard MHQA datasets. First, whether using LLaMA3-8B-Instruct or gpt-3.5-turbo-instruct as the LLM backbones, itera-tive RAG methods, especially IRCoT, significantly outperform the single-round RA-CoT, confirm-ing that multiple retrievals can indeed enhance the multi-step reasoning capabilities of LLMs for the open-domain MHQA task. Second, two emerging restructuring-based methods, i.e., SG-Prompt and EAR-CoT, conduct one-time retrieval (due to the intricacy of restructuring generations), and make efforts to reorganize the retrieved passages. Despite retrieving only once, these two approaches still achieve comparable performance to the iterative IRCoT, indicating the rationality of leveraging the underlying knowledge structure to enhance LLM inference. Beyond all these methods, our paradigm takes a new perspective of explicit knowledge tracing to seamlessly integrate knowledge structuring with multi-step reasoning. One can observe that KNOWTRACE consistently surpasses the baselines on both evaluation metrics (i.e., EM and F1) across all three datasets. For example, compared with IRCoT and ERA-CoT, when LLaMA3-8B-Instruct is selected as the base LLMs, our KNOW-TRACE achieves approximately $5.3\%$ and $4.3\%$ average absolute EM gains, respectively. When the base LLMs are switched to gpt-3.5-turbo-instruct, the gains increase to $6.7\%$ and $4.5\%$ accordingly. Such advanced performance demonstrates the superiority of our paradigm in multi-step inference, and we further explore the potential of self-improvement in the next part.

## 4.3 EFFECTIVENESS OF KNOWLEDGE BACKTRACING FOR SELF-IMPROVEMENT

In this part, we investigate the effectiveness of backtracing-guided self-training for KNOWTRACE. We refer to the improved version as KNOWTRACE*. For comparison, we also use the vanilla self-improvement method (Zelikman et al., 2022) to derive a *non-backtracing* version. On the one hand, we compare their inference performance (EM) in each self-training iteration. On the other hand, we also consider such a statistical indicator: during data collection in each self-training iteration, since the backtracing mechanism can naturally identify which LLM generations in the positive trajectories should be filtered, we then calculate the ratio of the tokens that should be filtered to all output tokens. We refer to this ratio as FA (Filtered-to-All). A larger FA means that the collected positive trajecto-ries contain more irrelevant generations, thereby indicating inferior quality of the finetuning dataset. We use this ratio to measure the proportion of noisy data in each self-training iteration.

Fig. 2 shows the EM results and FA ratios of the two self-trained KNOWTRACE versions. In terms of the inference performance (a-c), we observe that the backtracing-guided KNOWTRACE* can achieve performance gains during self-training until it reaches saturation, while the vanilla self-improvement process causes performance degradation instead, which we attribute to its disregard for useless LLM generations in the collected correct trajectories. Unlike the one-time generation tasks in the original

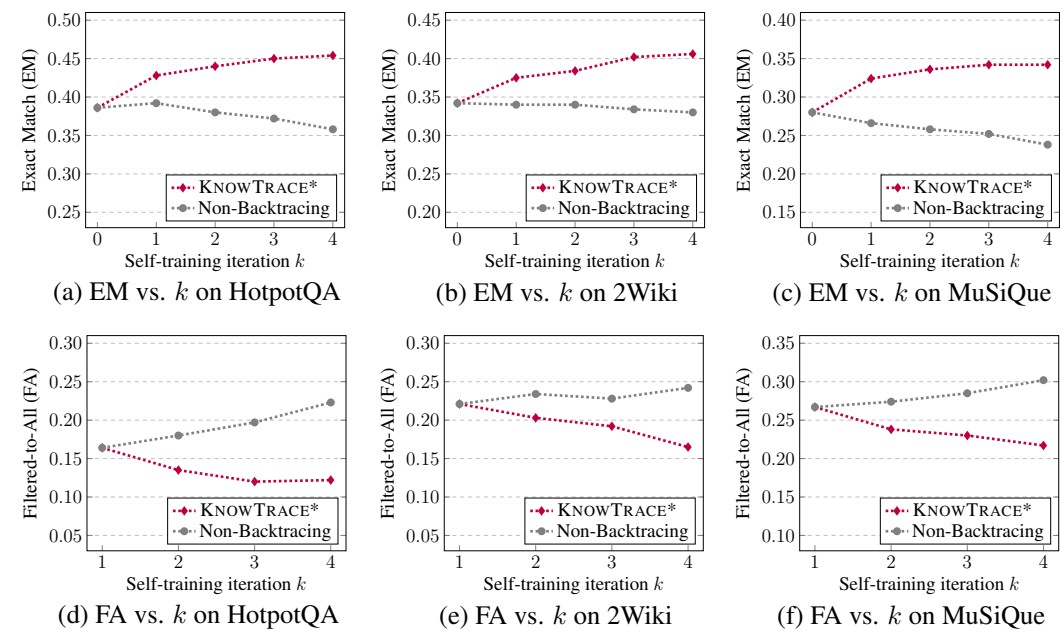

Figure 2: EM results (a-c) and FA ratios (d-f) in each self-improvement iteration. KNOWTRACE* is finetuned under the guidance of our knowledge backtracing mechanism, while Non-Backtracing is trained with the vanilla self-improvement process (Zelikman et al., 2022). We select the open-source `LLaMA3-8B-Instruct` as the base LLM for finetuning. The FA ratios measures the proportion of irrelevant generations that should be filtered out, as described in Section 4.3.

paper (Zelikman et al., 2022), the multi-step reasoning process exhibits such complexity: a correct reasoning trajectory may still contain multiple useless LLM generations. We use the FA indicator to reveal this phenomenon from a statistical perspective. As shown in Fig. 2(d)-2(f), for the fist data collection (i.e., $k = 1$), there are more than $10\%$ (even $26.7\%$) irrelevant generations in the collected positive trajectories. Indiscriminately finetuning on such data results in a negative synergistic effect: the noisy data impairs the generation quality, which in turn causes the generated correct trajectories to contain more noise. Our knowledge backtracing mechanism is capable of identifying this type of noise, thus enabling effective self-improvement for KNOWTRACE. We highlight that such reflective mechanism is naturally built upon the self-organized KGs, thus further confirming the rationality of our design perspective of explicit knowledge tracing.

## 4.4 EFFECT OF DIFFERENT CONFIGURATIONS

**Retrieval methods.** We demonstrate the compatibility of KNOWTRACE across different retrievers. Specifically, in addition to BM25 used in Table 1, we further conduct experiments with two retrieval methods: DPR and Contriever. Table 2 reports the EM results of KNOWTRACE and two advanced baselines (i.e., IRCoT and ERA-CoT) under these three retrievers. One can observe that our proposal consistently surpasses both baselines on all the datasets, regardless of the type of retrieval methods. Such superior performance confirms the general applicability of our approach on various retrievers.

**Number of retrieved passages.** We further investigate the effect of the number of retrieved passages (i.e., $N$). Fig. 3 shows the EM results of our models and one restructuring-based baseline (i.e., ERA-CoT) with varying $N$ on HotpotQA dataset. Based on this figure, one can observe that our proposals consistently surpass the baseline by a clear margin across all the values of $N$. Moreover, ERA-CoT exhibits performance degradation when $N$ is relatively large (i.e., more than 20), which we attribute to the lack of explicit reasoning guidance during the intricate restructuring process. In contrast, the performance of both KNOWTRACE versions improves until saturation as we enlarge the value of $N$. This stronger and more stable performance demonstrates the effectiveness of seamlessly integrating reasoning and structuring within our paradigm.

Table 2: EM results for the models using three different retrieval methods. We commonly select $N = 5$ most relevant passages for each retrieval, and set `LLaMA3-8B-Instruct` as backbones.

| Method | HotpotQA | | | 2Wiki | | | MusiQue | | |
|---|---|---|---|---|---|---|---|---|---|
| | BM25 | DPR | Contriver | BM25 | DPR | Contriver | BM25 | DPR | Contriver |
| IRCoT | .324 | .252 | .332 | .286 | .214 | .280 | .240 | .126 | .252 |
| ERA-CoT | .344 | .286 | .348 | .294 | .220 | .312 | .242 | .134 | .246 |
| KNOWTRACE | **.386** | **.320** | **.398** | **.342** | **.246** | **.354** | **.280** | **.176** | **.288** |

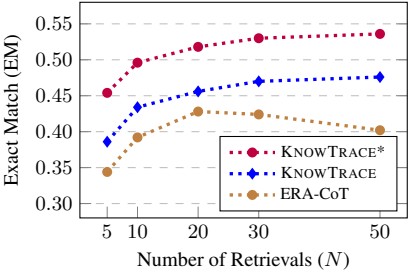

Figure 3: EM with varying $N$ on HotpotQA under `LLaMA3-8B-Instruct` and BM25.

Table 3: EM/F1 results for KNOWTRACE with three different knowledge prompting strategies.

| Strategy | HotpotQA | 2Wiki | MusiQue |
|---|---|---|---|
| *KG-to-Triplets* | .386/.479 | .342/.403 | .280/.387 |
| *KG-to-Paths* | .382/.465 | .334/.392 | .286/.398 |
| *KG-to-Text* | .376/.471 | .320/.386 | .274/.383 |

**Knowledge prompting strategies.** Since this work highlights the significance of explicit knowledge structures (i.e., KGs) for LLM inference, a meaningful concern lies in how to incorporate KGs into LLM prompts. Here, we explore three types of prompting strategies, including *KG-to-Triplets*, *KG-to-Paths*, and *KG-to-Text*, which correspond to elementary triplets, connected paths, and rewritten natural language, respectively, as described in Section 3.2. Table 3 presents the EM/F1 results of our paradigm with these three strategies. For fair comparisons, we utilize `LLaMA3-8B-Instruct` as the base LLM and BM25 as the retriever. One the one hand, the simplest *KG-to-Triplets* works well, while organizing independent triplets into paths (i.e., *KG-to-Paths*) does not lead to consistent gains. We find that the path extraction process typically duplicates some triplets, which may distract LLM inference (Shi et al., 2023; Ji et al., 2023). On the other hand, converting KGs back into natural text with LLM (i.e., *KG-to-Text*) also results in inferior performance, which we attribute to the absence of priori structural templates in the prompts. For example, one can directly specify that the contexts take the form of (*subject*, *relation*, *object*) in the triplet prompting strategy. Overall, *KG-to-Triplets* exhibits the dual advantages of simplicity and effectiveness, as it can offer a priori structural template without bringing duplicate information, making it the main choice for our experiments.

## 5 DISCUSSION AND CONCLUSION

**Limitations.** Although our paradigm showcases advantages in both multi-step reasoning and self-taught finetuning, there are still two major limitations in this work. First, despite its effectiveness in MHQA task, the applicability of our explicit knowledge tracing in other complex scenarios, such as mathematics and decision-making tasks, is unexplored. Second, KNOWTRACE can leverage the designed backtracing mechanism to foster self-taught finetuning, but how to proactively correct erroneous trajectories without finetuning remains an open challenge in our paradigm. We expect future studies to mitigate these issues.

**Conclusion.** In this work, we introduce KNOWTRACE, a simple yet effective RAG paradigm to enhance the multi-step reasoning capabilities of LLMs for more advanced MHQA performance. Our design idea is to seamlessly integrate the knowledge structuring with the multi-step reasoning from the perspective of explicit knowledge tracing. Benefiting from this perspective, our KNOWTRACE not only acquires question-specific KGs to facilitate inference, but also naturally inspires a reflective backtracing mechanism to stimulate self-improvement. Extensive experiments over three standard MHQA benchmarks comprehensively demonstrate the superiority of our proposals. Under different configurations of LLMs and retrieval models, our paradigm consistently outperforms a series of existing RAG approaches, and the backtracing-guided finetuning further elevates the overall performance, thereby showcasing the rationality of our perspective of explicit knowledge tracing.

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

## A  GLOSSARY OF SYMBOLS

Table 4: Glossary of variables and symbols used in this paper.

| Symbol | Description |
|:---:|:---|
| $q$ | Input question |
| $t$ | Chain of thought in the final LLM generation |
| $a$ | Predicted answer in the final LLM generation |
| $\hat{a}$ | Ground truth answer |
| $M$ | Base LLM |
| $C$ | Large-scale corpus |
| $I_{\mathrm{exp}}$ | Prompt template for exploration phase |
| $I_{\mathrm{com}}$ | Prompt template for completion phase |
| $\mathcal{G}_q$ | $q$-specific KG acquired by our KNOWTRACE |
| $\mathcal{S}_q$ | Subgraph of $\mathcal{G}_q$ that exactly supports the prediction (i.e., graph rationales) |
| $\mathcal{D}$ | Labeled dataset for the self-taught finetuning process of KNOWTRACE |
| $\mathcal{P}$ | Output set of exploration phase |
| $\mathcal{C}^N_{(e_i, r_i)}$ | $N$ relevant passages retrieved for $(e_i, r_i)$ |
| $\mathcal{T}_{(e_i, r_i)}$ | Output set of completion phase for entity-relation pair $(e_i, r_i)$ |
| FLAG | Boolean identifier indicating whether the acquired knowledge is sufficient |

## B  DATASETS

We use the following three widely-used multi-hop question answering datasets for evaluation:

**HotpotQA.** This is a comprehensive dataset collected from the English Wikipedia, comprising approximately 113k crowd-sourced questions. The unique characteristic of HotpotQA lies in its construction, which requires answers to be derived from the introductory paragraphs of two distinct Wikipedia articles. For each question, the dataset includes the corresponding gold paragraphs from these articles, alongside a curated list of sentences identified by crowdworkers as critical supporting evidence necessary to accurately resolve the query. Note that our evaluation is conducted under the open-domain setting (Trivedi et al., 2023), and thus does not use these gold information.

**2WikiMultihopQA (2Wiki).** This dataset consists of complex 2-hop questions that require either compositional reasoning or comparative analysis. Both structured and unstructured information from Wikipedia and Wikidata are combined for data construction. In terms of difficulty, 2WikiMultihopQA is challenging for multi-hop models and it ensures that multi-hop reasoning is required.

**MuSiQue.** The multi-hop questions in this dataset is constructed by carefully selecting and composing single-hop questions obtained from a large collection of single-hop questions. In terms of difficulty, MuSiQue is a more challenging dataset, since it contains 2 to 4 hop questions.

## C  BASELINES

**RA-CoT (Wei et al., 2022).** This is the simplest approach, which conducts one-time retrieval with the input question as the query, and use the retrieved text to guide LLM to perform CoT reasoning.

**ReAct (Yao et al., 2023).** This approach integrates reasoning, action, and observation steps in an iterative process until a final answer is reached. Actions in this process include generating queries to search for relevant information or concluding with a final answer. Observations are formed by concatenating the results from these actions and serve as inputs for subsequent reasoning steps.

**Self-Ask (Press et al., 2023).** This work adopts an iterative approach to break down complex questions into simpler sub-questions. At each iteration, sub-questions are generated based on the current stage of reasoning, followed by retrieving relevant information and answering these sub-questions. This process continues until the answer is finalized.

**Iter-RetGen (Shao et al., 2023).** This work leverages the LLM output from the previous iteration as the query to retrieve more relevant knowledge. It conducts retrievals by concatenating the output from the previous iteration with the original question.

**SG-Prompt (Li & Du, 2023).** This work first constructs a semantic graph structures through information extraction from all retrieved text, and then leverages this symbolic information (including entities and semantic relations) to enhance LLM's inference process.

## D  PROMPTS

---

**Prompt Template for Knowledge Exploration**

```
Given a question that requires multi-step retrieval to collect
↪   necessary knowledge triplets and offer the final answer, you
↪   are an advanced knowledge reasoner and retrieval facilitator.
↪   The knowledge triplets that you collected in previous steps
↪   take the form of (subject entity; relation; object entity).

In this step, you should first carefully determine whether the
↪   collected knowledge triplets are sufficient for you to offer
↪   the final answer. Don't answer with uncertainty. Please
↪   strictly use the following judgment template:
Whether the given knowledge triplets are sufficient for answering:
↪   Yes or No

If Yes, then think and offer the final answer based on the
↪   collected knowledge triplets. Please strictly use the
↪   following inference template:
Thought: think step by step to reason out the final answer
Answer: the final answer

If No, then provide a high-quality concrete guidance for the
↪   retrieval step to collect more necessary knowledge triplets.
↪   You should first provide a set of entities that need futher
↪   retrieval in the retrieval step, and then propose a detailed
↪   and concrete relation guidance for each entity to reflect
↪   which aspect of knowledge related to this entity you want to
↪   retrieve. Be sure to only provide the relation guidance for
↪   necessary knowledge. Please strictly use the following
↪   template:
Retrieval Guidance:
- Entity name 1: propose a detailed and concrete retrieval
↪   guidance for this entity
- Entity name 2: propose a detailed and concrete retrieval
↪   guidance for this entity
- ...

{4-Shot Examples}

Question: {Question}
Knowledge triplets collected in previous steps: {Triplets}
```

---

Figure 4: Prompt Template for Knowledge Exploration.

```
Prompt Template for Knowledge Completion

    Given a set of documents and a specified entity with a knowledge
    ↪   guidance on the entity-related knowledge, you are an advanced
    ↪   relevant knowledge extractor. According to the provided
    ↪   knowledge guidance, you should extract sufficient information
    ↪   from the documents to construct the structured knowledge
    ↪   triplets that are related to the input entity. The constructed
    ↪   knowledge triplets must take the complete form of (subject
    ↪   entity; relation; object entity), in which the relation must
    ↪   be detailed and concrete. You must provide the knowledge
    ↪   triplets without any vague expressions such as "not found" or
    ↪   "N/A". Use newline characters as separators between multiple
    ↪   knowledge triplets. Feel free to ignore irrelevant knowledge
    ↪   in the documents.

    The input entity with knowledge guidance are organized as follows:
    - Input entity name: a knowledge guidance for this entity

    Please strictly use the following triple template, and do not
    ↪   provide any unnecessary explanations or notes.
    (subject entity; relation; object entity)\n(subject entity;
    ↪   relation; object entity)\n...

    {4-Shot Examples}

    Documents:
    {Documents}
    Input Entity with Knowledge Guidance:
    {Entities-Relation Guidance}
    Structured Knowledge Triple(s):
```

Figure 5: Prompt Template for Knowledge Completion.

# E    COST ANALYSIS OF KNOWTRACE

In this section, we include a detailed cost and latency analysis for KNOWTRACE and two representative baselines (i.e., IRCoT and ERA-CoT). The statistics are summarized in Table 5.

Table 5: Cost statistics of KNOWTRACE and two representative baselines. #Iter is the average number of inference iterations per question; #Tok is the average number of retrieved tokens processed by LLMs per question; #Time is the average inference time per question.

| Method | HotpotQA | | | 2Wiki | | | MusiQue | | |
|---|---|---|---|---|---|---|---|---|---|
| | #Iter | #Tok | #Time | #Iter | #Tok | #Time | #Iter | #Tok | #Time |
| IRCoT | 3.2 | 1.2 | 5 | 2.8 | 1.5 | 6 | 4.6 | 1.7 | 8 |
| ERA-CoT | 1.0 | 2.1 | 13 | 1.0 | 2.3 | 15 | 1.0 | 2.4 | 16 |
| KNOWTRACE | 2.5 | 1.4 | 6 | 2.4 | 1.5 | 6 | 3.8 | 1.8 | 9 |

Compared to the iterative baseline IRCoT, KNOWTRACE requires fewer iterations on average, since it can explore multiple expansion directions based on the current knowledge at each iteration, rather than solely relying on a single chain of thought. This allows KNOWTRACE to acquire more relevant knowledge in each iteration, reducing the overall number of iterations required.

For the restructuring-based baseline ERA-CoT, although it is a non-iterative RAG approach (#Iter = 1.0), its restructuring process involves 5 LLM-driven steps (entity extraction, relation extraction, relation inference, discrimination, and question answering) for every input question. These steps are inherently non-parallelizable and all require retrieved passages to be included in the LLM prompts.

Therefore, the resturcturing operations in ERA-CoT incur significantly higher inference time cost than both IRCoT and our KNOWTRACE.

Overall, beyond the iterative and restructuring-based baselines, KNOWTRACE seamlessly integrates knowledge structuring with multi-step reasoning, enhancing inference performance without sacrificing the efficiency. In other words, KNOWTRACE achieves a favorable balance of computational cost and multi-step reasoning capability compared to both iterative and restructuring-based baselines.

## F    EXAMPLE OF KNOWTRACE INFERENCE AND BACKTRACING

To better illustrate the workflow of our KNOWTRACE framework, we present a succinct example of its inference and backtracing process, complementing the high-level presentation in Figure 1.

Table 6: A succinct example of KNOWTRACE inference from the MuSiQue dataset.

---

**Question:** Where was the person who wrote about the rioting being a dividing factor in Birmingham educated?

**[Iteration 1]**

```
# Knowledge Memory: None
# Knowledge Exploration
Whether the collected knowledge triplets are sufficient: No
What entity-relation pairs to retrieve in the current step:
- The rioting being a dividing factor in Birmingham: Find out who wrote
↪   about this topic.
- Birmingham: Find out where Birmingham is located.
# Retrieve Relevant Textual Passages From Wikipedia
# Knowledge Completion
(James Watt, wrote, the rioting being a dividing factor in Birmingham);
(James Watt, is, an industrialist);
(Birmingham, is located in, the West Midlands region of England)
```

**[Iteration 2]**

```
# Knowledge Memory: The Knowledge Triplets Acquired in Iteration 1
# Knowledge Exploration
Whether the collected knowledge triplets are sufficient: No
What entity-relation pairs to retrieve in the current step:
- James Watt: Find out which school James Watt attended.
# Retrieve Textual Passages From Wikipedia
# Knowledge Completion
(James Watt, was educated at, University of Glasgow)
```

**[Iteration 3]**

```
# Knowledge Memory: The Knowledge Triplets Acquired in Iteration 1 and 2
# Knowledge Exploration
Whether the collected knowledge triplets are sufficient: Yes
Thought: James Watt wrote about the rioting being a dividing factor in
↪   Birmingham. He was educated at the University of Glasgow.
Answer: University of Glasgow
```

---

The inference example of KNOWTRACE is shown in Table 6. According to the transparent structure traced out in this example, one can naturally backtrace from the answer entity *University of Glasgow* to identify the following evidence subgraph: *(James Watt, wrote, the rioting being a dividing factor in Birmingham)*; *(James Watt, was educated at, University of Glasgow)*. In this way, our framework naturally allows for filtering out unavailing exploration (e.g., "- *Birmingham: Find out where Birmingham is located*") and extraneous completion (e.g., *(James Watt, is, an industrialist)*) from the LLM generations, thereby producing higher-quality reasoning rationales for the self-improvement.

## G    DISCUSSION ON MORE RELATED WORKS

Several recent works also leverage structured information to enhance the training of language models or guide their reasoning processes. (Wang et al., 2023b) and (Misra et al., 2023) focus on con-

structing masked knowledge structures as training data for (pre-)training language models, aiming to imbue the models with structural reasoning capabilities. Specifically, they construct training datasets by first restructuring Wikipedia documents and then masking specific (predefined or random-walk-generated) entities within the structures. In contrast, our method does not rely on such structural pretraining or dataset construction, but instead operates directly on unstructured text, actively tracing relevant knowledge in the form of triplets during multi-step inference.

GE-Reasoning (Park et al., 2023) and Semi-Structured CoT (Su et al., 2023) focus on parsing input questions into masked structured chains and subsequently rely either on existing external knowledge graphs to fill missing triplets or rewrite missing triplets as natural language queries to retrieve answers from external text databases. However, such approaches heavily depend on the accuracy of the initial parsing—errors at this stage can propagate—thereby necessitating careful filtering and consistency operations (Su et al., 2023). In contrast, KNOWTRACE adopts a more flexible perspective of adaptively tracing knowledge triplets during the multi-step reasoning process, rather than solely relying on the one-time parsing of the input question. This adaptive exploration can reduce error propagation and enhance robustness.

CoK (Wang et al., 2023a) retrieves candidate knowledge triplets from a pre-constructed KG and combines them with human annotations, aiming to design effective exemplars that induce fact generation capabilities of LLMs. In contrast, our work pursues a different objective, i.e., tracing and expanding structured knowledge directly from unstructured text during multi-step reasoning process to enhance the multi-step reasoning capabilities of LLMs.

Overall, our KNOWTRACE framework actively traces knowledge triplets relevant to the input question during multi-step reasoning process. Such a perspective enables more flexible LLM inference and does not require additional structural training or one-time parsing of the input. The progressive expansion of structured knowledge memory in our KNOWTRACE framework not only enhances LLM inference, but also provides a transparent record of the reasoning Procedure. This transparency allows for the natural backtracing mechanism to distill higher-quality rationales, which can further be leveraged for post-training (e.g., self-improvement). The proposed framework is orthogonal to the above techniques, and one can integrate them to further enhance the reasoning capabilities of LLMs. For instance, KNOWTRACE could use models pre-trained with structural reasoning as the backbone or incorporate pre-parsed question structures to assist in the knowledge exploration phase.

