# OpenReview forum: "KnowTrace: Explicit Knowledge Tracing for Structured Retrieval-Augmented Generation"
_ICLR.cc/2025/Conference — Submitted to ICLR 2025_

### Official Review · Reviewer_8nP3 · 2024-10-30

**Soundness:** 3
**Presentation:** 3
**Contribution:** 3
**Rating:** 6
**Confidence:** 4

**Summary:**

Previously, most iterative RAG methods accumulate all retrieved passages into LLM prompts, creating challenges for handling long context and unstructured text. A more helpful way would be to develop specific structures from these passages for the LLM to better understand. To seamlessly incorporate the informative restructuring process into the iterative RAG for higher-quality multi-step reasoning, the authors propose KnowTrace, which coherently traces out question-specific knowledge structures to bolster multi-step reasoning.
Specifically, KnowTrace alternates between knowledge exploration and knowledge completion. During Knowledge Exploration, the LLM determines a set of entities and respective relations based on the current KG. During Knowledge Completion, the LLM fills in the entities based on the retrieved passages. Moreover, the authors illustrate a backtracking process that maps out a subgraph that contribute to the final prediction and use this to distill high-quality rationales. The mechanism is incorporated into self-improvement.
In the experiments, KnowTrace consistently outperforms other baselines. With backtracking, KnowTrace's performance improves in each iteration.

**Strengths:**

1. The method motivation is clear, which is restructuring the retrieved passages to facilitate better reasoning.
2. KnowTrace demonstrates stable improvement across all datasets and setups. The authors conducted ablations in knowledge prompting strategies, number of retrievals, etc. Figure 3 is particularly interesting, which shows that KnowTrace can improve scenarios with a large number of retrievals, suggesting better abilities to organize important information.
3. The self-improvement loop shows promising results on scaling such RAG methods.

**Weaknesses:**

No specific weaknesses.

**Questions:**

1. It seems that this self-improvement training and the inference steps could incur some overhead. Could you include a cost analysis?
2. The retrieved passages are in the form of free text, which is then transformed into knowledge triplets with the LLM. Would it be possible to directly retrieve from some KG?
3. For the back-tracing step, if the data is collected with a more powerful backbone model, would the quality be improved? Would the better-quality data contribute to more effective training for improvement?

---

> ### Author Response · Authors · 2024-11-18
> **Response to Reviewer 8nP3 (1/2)**
>
> We sincerely appreciate your careful reading and positive comments on our paper. Below, we provide a detailed response to your concerns and questions.
>
> > **Q1: Include a cost analysis of KnowTrace.**
>
> Thanks for raising this important point. We follow your suggestion to include **a detailed cost and latency analysis** for our KnowTrace and two representative baselines (i.e., IRCoT and ERA-CoT). The statistics are summarized as follows:
>
> | Dataset | Method                                                 | #Iteration  | #Token (k)  | #Time (s) |
> | :------------ | :------------------------------------------------------------ | :------------: | :------------: | :----------: |
> | HotpotQA | IRCoT  | 3.2 | 1.2 |    5     |
> | | ERA-CoT                      |     1.0    | 2.1  |    13     |
> | | KnowTrace (ours)                                               |     2.5   | 1.4   |  6  |
> | 2Wiki | IRCoT  | 2.8 | 1.5 |    6      |
> | | ERA-CoT                      |     1.0    | 2.3  |    15      |
> | | KnowTrace (ours)                                               |     2.4   | 1.5   |  6  |
> | MuSiQue | IRCoT  | 4.6 | 1.7 |    8      |
> | | ERA-CoT                      |     1.0    | 2.4  |    16      |
> | | KnowTrace (ours)                                               |     3.8   | 1.8   |  9  |
>
> **#Iteration**: Average number of inference iterations per question
> **#Token**: Average number of tokens processed by LLMs per question
> **#Time**: Average inference time per question
>
> We can observe that:
> - Compared to the iterative baseline IRCoT, KnowTrace requires fewer iterations on average, since it can explore **multiple expansion directions** based on the current knowledge structure at each iteration, rather than solely relying on a single chain of thought. This allows KnowTrace to acquire more relevant knowledge in each iteration, reducing the overall number of iterations required.
> - For the restructuring-based baseline ERA-CoT, although it is a non-iterative approach (#Iteration = 1.0), its restructuring process involves 5 LLM-driven steps (entity extraction, relation extraction, relation inference, discrimination, and question answering) for every input question. **These steps are inherently non-parallelizable and all require retrieved passages to be included in the LLM prompts.** Therefore, the resturcturing operations in ERA-CoT incur significantly higher inference time cost than both IRCoT and our KnowTrace.
> - Overall, beyond the iterative and restructuring-based baselines, KnowTrace seamlessly integrates knowledge structuring with multi-step reasoning, enhancing inference performance **without sacrificing the efficiency**. In other words, **KnowTrace achieves a favorable balance of computational cost and multi-step reasoning capability compared to both iterative and restructuring-based baselines.**
>
> At the same time, we would like to respectfully clarify that the backtracing mechanism naturally leverages the knowledge structures organized during KnowTrace inference **without additional LLM calls**. This mechanism produces high-quality rationales for self-improvement training, whose cost aligns with standard parameter-efficient fine-tuning (approximately 2–3 hours on a single NVIDIA A100 GPU).
>
> > **Q2: Would KnowTrace be possible to directly retrieve from external KG?**
>
> **Yes, KnowTrace is able to leverage external KG to enhance LLM inference.** Specifically, after determining the entity-relation pair $(e, r)$ to explore in the current iteration, KnowTrace can naturally adopt a structured retrieval approach: first, select the top-$m$ entities from the external KG that are most similar to $e$; next, among the relations associated with these entities, select the top-$n$ relations most similar to $r$; the triplets corresponding to these entities and relations in the external KG are then traced out for this iteration.
>
> We validate the effectiveness of this approach on a standard multi-hop KGQA dataset (CWQ [1]). The EM results are shown as follows, demonstrating that KnowTrace can effectively acquire relevant knowledge triplets from external KG to enhance LLM inference.
> | Method                                                 | CWQ (EM) |
> | :------------------------------------------------------------ | :------------: |
> | Direct IO (LLaMA3-8B-Instruct)  | 0.352 |
> | CoT (LLaMA3-8B-Instruct)  | 0.394 |
> | KnowTrace (LLaMA3-8B-Instruct)  |     0.506   |

---

> > ### Author Response · Authors · 2024-11-18
> > **Response to Reviewer 8nP3 (2/2)**
> >
> > > **Q3: For the backtracing step, would using a more powerful backbone model improve the data quality and contribute to more effective training?**
> >
> > Regarding your interest in combining the backtracing mechanism with a more powerful backbone model, we explore this idea on HotpotQA dataset. Specifically, we use gpt-3.5-turbo-instruct as the backbone model to collect data, and apply the backtracing mechanism to filter out irrelevant generations. The distilled generations are then used to finetune the basic KnowTrace (using LLaMA3-8B-Instruct as the backbone). The EM results and the noise filtering ratio are shown below. One can observe that the data collected with a more powerful backbone provides positive supervisory signals for finetuning. Moreover, our backtracing mechanism further enhances the data quality (filtering approximately 9% of noisy generations on average), leading to superior finetuning performance. **This analysis demonstrates the potential of leveraging more powerful backbones in conjunction with the backtracing mechanism to improve training data quality and enhance KnowTrace performance**.
> >
> > | Basic Backbone |   Backbone for data collection | Average Filtering Ratio of Backtracing | EM |
> > | :------------------------------------------------------------: | :------------: | :------------: | :------------: |
> > | LLaMA3-8B-Instruct | Not Used  | Not Used | 0.386 |
> > | LLaMA3-8B-Instruct | LLaMA3-8B-Instruct  | 14% | 0.452 |
> > | LLaMA3-8B-Instruct | gpt-3.5-turbo-instruct  | Not Used | 0.466 |
> > | LLaMA3-8B-Instruct | gpt-3.5-turbo-instruct  | 9% | 0.498 |
> >
> >
> > **References:**
> > [1] The Web as a Knowledge-base for Answering Complex Questions.

---

> > > ### Comment · Reviewer_8nP3 · 2024-11-25
> > > **Reply**
> > >
> > > Thank you for the additional clarification and experiments. I believe the original score is already good and would like to maintain my score.

---

### Official Review · Reviewer_Tjwb · 2024-11-04

**Soundness:** 3
**Presentation:** 3
**Contribution:** 2
**Rating:** 6
**Confidence:** 5

**Summary:**

This paper focuses on using large language models (LLMs) for knowledge-intensive, multi-hop question answering. Specifically, it introduces an approach to iteratively store and combine retrieved relevant knowledge in the form of knowledge graph triples, ultimately using this structured information for answering questions. The structured nature of the reasoning process is also leveraged to filter the generated reasoning paths, and the model is also fine-tuned based on these filtered reasoning chains.

**Strengths:**

- The paper is well-written, clear, and easy to follow. The proposed method is straightforward, intuitive, and easy to implement.
- It is innovative that the paper leverages the structured nature of reasoning paths to filter and refine generated trajectories for model training.
- The method demonstrates strong empirical performance across multiple datasets compared to the baseline methods.

**Weaknesses:**

My main concern is with the novelty of this approach. Representing knowledge or reasoning processes in a structured format has been explored in several prior works [1, 2, 3, 4, 5], many of which were also tested on similar benchmark datasets. These works considered not only structured representations but also integrated unstructured knowledge to include information that may not fit neatly into a knowledge graph. The core idea is thus similar. The FLAG mechanism for knowledge exploration here also resembles the self-ask approach (also mentioned in this paper), where models automatically stop querying and generate an answer. It would be helpful for the authors to provide a more detailed comparison with these works to highlight the unique contributions of this paper.

[1] Unifying Structure Reasoning and Language Model Pre-training for Complex Reasoning
[2] Graph-Guided Reasoning for Multi-Hop Question Answering in Large Language Models
[3] Semi-Structured Chain-of-Thought: Integrating Multiple Sources of Knowledge for Improved Language Model Reasoning
[4] Triggering Multi-Hop Reasoning for Question Answering in Language Models using Soft Prompts and Random Walks
[5] Boosting Language Models Reasoning with Chain-of-Knowledge Prompting

**Questions:**

In datasets like HotpotQA and Musique, many supporting facts may be difficult to represent solely in the form of triples. How did you address such cases?

---

> ### Author Response · Authors · 2024-11-18
> **Response to Reviewer Tjwb (1/2)**
>
> We sincerely appreciate your careful reading and valuable comments on our paper. Below, we provide a detailed response to your concerns and questions.
>
> > **Q1: Compare KnowTrace with more prior works, and highlight the unique contributions.**
>
> We understand your concern and appreciate the opportunity to clarify the novelty of our work compared to [1–5]. Below, we highlight the key distinctions and our unique contributions.
>
> **Comparison with [1] and [2]:** These two works focus on constructing masked knowledge structures as training data for pretraining (or finetuning) language models, aiming to imbue the models with structural reasoning capabilities. Specifically, they construct training datasets by first restructuring Wikipedia documents and then masking specific (predefined or random-walk-generated) entities within the structures. In contrast, our method **does not rely on such structural pretraining or dataset construction**, but instead operates directly on unstructured text, actively tracing relevant knowledge in the form of triplets during multi-step inference.
>
> **Comparison with [3] and [4]:** These two works focus on parsing input questions into masked structured chains and subsequently rely either on existing external knowledge graphs to fill missing triplets or rewrite missing triplets as natural language queries to retrieve answers from external text databases. However, such approaches **heavily depend on the accuracy of the initial parsing**---errors at this stage can propagate---thereby necessitating careful filtering and consistency operations [4]. In contrast, KnowTrace adopts a more **flexible** perspective of adaptively tracing knowledge triplets during the multi-step reasoning process, rather than solely relying on the one-time parsing of the input question. This adaptive exploration can reduce error propagation and enhance robustness.
>
> **Comparison with [5]:** This work retrieves candidate triplets from a pre-constructed KG and combines them with human annotations, aiming to design effective exemplars that induce fact generation capabilities of LLMs. In contrast, our work **pursues a different objective**, i.e., tracing and expanding structured knowledge directly from unstructured text during multi-step reasoning process to enhance the multi-step reasoning capabilities of LLMs.
>
> Overall, **the unique contributions of our work** are summarized as follows:
>
> **Flexible Knowledge Exploration and Structuring.** KnowTrace actively traces knowledge triplets relevant to the input question during multi-step reasoning process. Such a perspective enables more flexible LLM inference and does not require additional structural training or one-time parsing of the input.
>
> **Transparent Reasoning Procedure.** The progressive expansion of structured knowledge memory in our KnowTrace framework not only enhances LLM inference, but also provides a transparent record of the reasoning Procedure. This transparency allows for the natural backtracing mechanism to distill higher-quality rationales, which can further be leveraged for post-training (e.g., self-improvement).
>
> **Complementary to the Prior Works.** The proposed framework is orthogonal to the techniques in [1–5], and one can integrate them to further enhance the reasoning capabilities of LLMs. For instance, KnowTrace could use models pre-trained with structural reasoning (as in [1], [2]) as the backbone or incorporate pre-parsed question structures (as in [3], [4]) to assist in the knowledge exploration phase.
>
> We hope these detailed comparisons address your concern, and will incorporate these discussions into the revision for better positioning our proposal relative to the prior works [1-5].

---

> ### Author Response · Authors · 2024-11-18
> **Response to Reviewer Tjwb (2/2)**
>
> > **Q2: How to address the supporting facts that may be difficult to represent solely in the form of triplets?**
>
> Indeed, certain information, such as **temporal attributes, quantities, or qualitative descriptions**, may not be neatly expressed as the standard subject-predicate-object triplets in conventional knowledge graphs. To address this, KnowTrace adopts a more flexible approach by **allowing free-form textual descriptions within the structured triplets**. This enables the representation of nuanced relationships and attributes. For example, from "John moved to Paris before 2015", the free-form triplet could be: (John, moved to before 2015, Paris), embedding temporal attribute directly into the relation. This flexible representation allows KnowTrace to adaptively trace and structure knowledge even in cases where conventional triplet formats fall short. We believe this adaptability is key to handling the complexity of datasets like HotpotQA and MuSiQue effectively.
>
>
> **References:**
> [1] Unifying Structure Reasoning and Language Model Pre-training for Complex Reasoning
> [2] Triggering Multi-Hop Reasoning for Question Answering in Language Models using Soft Prompts and Random Walks
> [3] Graph-Guided Reasoning for Multi-Hop Question Answering in Large Language Models
> [4] Semi-Structured Chain-of-Thought: Integrating Multiple Sources of Knowledge for Improved Language Model Reasoning
> [5] Boosting Language Models Reasoning with Chain-of-Knowledge Prompting

---

> ### Author Response · Authors · 2024-11-25
> **A Gentle Reminder**
>
> Dear Reviewer Tjwb,
>
> Thank you very much for your time and insightful feedback. We have addressed your questions and comments with detailed clarifications. We would greatly appreciate it if you could reconsider your rating. Additionally, we are more than happy to answer any further questions you may have.
>
> Our responses can be outlined as follows:
> - First, we provided **a detailed clarification of the key distinctions between our framework and the prior works [1-5]**.
> - Second, we highlighted the unique contributions of our work, including
>     - **Flexible Knowledge Exploration and Structuring**
>     - **Transparent Reasoning Procedure for Backtracing**
>     - **Complementary to the Prior Works**
> - Third, we clarified that: to enable the representation of nuanced relationships and attributes, KnowTrace allows **free-form textual descriptions within the structured triplets**, which we believe is crucial for handling the complexity of datasets like HotpotQA and MuSiQue effectively.
>
> Best regards,
> Authors

---

> ### Comment · Reviewer_Tjwb · 2024-11-27
> **Response to Author**
>
> Thank you for your response. It addressed some of my concerns, and I will improve my score. Please consider adding this discussion to the final version.

---

### Official Review · Reviewer_ifaQ · 2024-11-06

**Soundness:** 2
**Presentation:** 2
**Contribution:** 2
**Rating:** 6
**Confidence:** 3

**Summary:**

The paper proposes KnowTrace, an iterative retrieval-augmented generation (RAG) framework for complex, multi-hop question-answering tasks. KnowTrace consists of two main phrases: *(1)* knowledge exploration, which takes the input question and an initial knowledge graph (KG) and provide guidance (entity-relation pairs to expand) for further querying the retrieval corpus. *(2)* knowledge completion, which takes entity-relation pairs and retrieved passages and output completed knowledge triplets for an enriched KG for the next iteration, before reaching a desired answer.
In addition, the paper also introduces an extended self-taught finetuning framework leveraging rationale data from KnowTrace's explicit knowledge trajectory in the process.

**Strengths:**

- The paper targets a challenging multi-hop QA setting in RAG and presents a new framework supported by strong empirical performance.

- The paper is generally well-written, well-motivated, and introduces a cleaver incorporation of knowledge graph operation for structured RAG.

- The paper includes multiple, comprehensive baselines to support the main experimental results of the proposed approach.

**Weaknesses:**

- It is not entirely clear between the difference of KnowTrace and iterative restructuring-based RAG. To the best of my understanding, the restructuring-based RAG approaches adopted as the baselines only involve one-time inference. However, considering that KnowTrace employs multiple iteration, it might be fair to also compare restructuring-based RAG approaches in an iterative setting.

- The presentation could be further improved for better clarity. In particular, it is relatively difficult for readers to imagine the inputs and outputs at different stages of KnowTrace with only the high-level conceptual framework (Figure ```1```). For instance, it might be hard to imagine the so called "guidance" provided by knowledge exploration. It would be greatly beneficial if an actual example of the input and generation results is accompanied in the paper.

- The self-taught finetuning KnowTrace$^*$ could be further enhanced with the iterative RAG baselines which also provide rationales (CoT) or other related works (e.g., InstructRAG [1]).

[1] Wei et al, *InstructRAG: Instructing Retrieval-Augmented Generation via Self-Synthesized Rationales*. 2024.

**Questions:**

- How many numbers of iterations does KnowTrace adopted in the experimental results of Table ```1```?

---

> ### Author Response · Authors · 2024-11-18
> **Response to Reviewer ifaQ (1/3)**
>
> We sincerely appreciate your careful reading and valuable comments on our paper. Below, we provide a detailed response to your concerns and questions.
>
> > **Q1: Compare KnowTrace with restructuring-based RAG approaches in the iterative setting.**
>
> We would first like to respectfully clarify that although the restructuring-based RAG approaches (such as ERA-CoT) are non-iterative, they typically **involve multiple intricate LLM-driven operations**. For instance, the restructuring process of ERA-CoT contains 5 LLM-driven steps (entity extraction, relation extraction, relation inference, discrimination, and question answering) for every input question. **These steps are inherently non-parallelizable and all require retrieved passages to be included in the LLM prompts**, thereby incurring even higher inference overhead than both iterative approaches (such as IRCoT) and our KnowTrace framework (you could find **a detailed cost analysis in our response to your Q4**).
>
> At the same time, we understand your concern about the **performance** (rather than inference overhead) comparison between iterative restructuring-based baselines and our KnowTrace. To address this, we extend ERA-CoT into the standard iterative setting for evaluation: iteratively propose new query for retrieval and restructure all retrieved passages until sufficient information is collected to derive the final answers. The EM/F1 and #Time (average inference time per question) results on HotpotQA dataset are summarized as follows:
>
> | Method                                                 | &nbsp;&nbsp;&nbsp;&nbsp;EM/F1 | #Time (s) |
> | :------------------------------------------------------------ | :------------: | :------------: |
> | IRCoT  | 0.324/0.425 | 5 |
> | One-Time ERA-CoT  | 0.344/0.435 | 13 |
> | Iterative ERA-CoT            |     0.370/0.452   | 29 |
> | KnowTrace (ours)             |     0.386/0.479   | 6 |
>
> (**#Time**: Average inference time per question)
>
> We can observe that iterative ERA-CoT outperforms its one-time counterpart but significantly increases the inference time per question. In contrast, **KnowTrace achieves more substantial performance gains without incurring high inference cost.** We attribute this to the favorable perspective of explicit knowledge tracing in our framework, which avoids the intricate process of indiscriminately restructuring all retrieved passages. Such restructuring process  used in (iterative) ERA-CoT may **retain extensive irrelevant information while overlooking knowledge critical to the input question**, which could negatively impact the subsequent query generation and the final answer inference [1].
>
> We hope this comparison clarifies the unique advantages of KnowTrace over both iterative and one-time restructuring-based RAG approaches.

---

> ### Author Response · Authors · 2024-11-18
> **Response to Reviewer ifaQ (2/3)**
>
> > **Q2: Better presentation with an actual example of KnowTrace.**
>
> Thanks for your valuable suggestion regarding the inclusion of an example to enhance the clarity of our framework. Following this suggestion, we provide a detailed example of KnowTrace inference and backtracing below, and will include it in our revision to complement the high-level presentation in Figure 1.
>
> **Example of KnowTrace Inference:**
> ```
> Input Question: Where was the person who wrote about the rioting being a dividing factor in Birmingham educated?
>
> [Iteration 1]
> ### Knowledge Memory: None
> ### Knowledge Exploration
> Whether the collected knowledge triplets are sufficient: No
> What entity-relation pairs to retrieve in the current step:
> - The rioting being a dividing factor in Birmingham: Find out who wrote about this topic.
> - Birmingham: Find out where Birmingham is located.
> ### Retrieve Textual Passages From Wikipedia
> ### Knowledge Completion
> (James Watt, wrote about, the rioting being a dividing factor in Birmingham);
> (James Watt, is, an industrialist);
> (Birmingham, is located in, the West Midlands region of England)
>
> [Iteration 2]
> ### Knowledge Memory: The Knowledge Triplets Acquired in Iteration 1
> ### Knowledge Exploration
> Whether the collected knowledge triplets are sufficient: No
> What entity-relation pairs to retrieve in the current step:
> - James Watt: Find out which school James Watt attended.
> ### Retrieve Textual Passages From Wikipedia
> ### Knowledge Completion
> (James Watt, was educated at, University of Glasgow)
>
> [Iteration 3]
> ### Knowledge Memory: The Knowledge Triplets Acquired in Iterations 1 and 2
> ### Knowledge Exploration
> Whether the collected knowledge triplets are sufficient: Yes
> Thought: James Watt wrote about the rioting being a dividing factor in Birmingham. He was educated at the University of Glasgow.
> Answer: University of Glasgow
> ```
>
> **Example of Backtracing Mechanism:**
> Based on the transparent knowledge structure traced by KnowTrace in the above example, we can naturally backtrace from the answer entity _University of Glasgow_ to identify the following evidence subgraph: **(James Watt, wrote about, the rioting being a dividing factor in Birmingham); (James Watt, was educated at, University of Glasgow)**. In this way, our framework naturally allows for filtering unavailing exploration (e.g., "- Birmingham: Find out where Birmingham is located") and extraneous completion (e.g., (James Watt, is, an industrialist)) from the LLM generations, thereby presenting higher-quality reasoning rationales for the self-improvement finetuning.
>
>
>
> > **Q3: Could KnowTrace\* be further enhanced with the iterative baselines or other related works such as InstructRAG [2]?**
>
>
> Yes, we agree with the potential to further enhance our framework via incorporating rationales or insights from the related works. For instance, the positive rationales (i.e., rationales leading to correct answers) generated by iterative RAG approaches could serve as auxiliary signals to guide KnowTrace* in producing higher-quality dataset for further finetuning.
>
> The concurrent work InstructRAG leverages additional LLM calls to generate rationales that explain how answers are derived from retrieved passages. Our framework could also naturally integrate this idea to augment the finetuning dataset, i.e., invoking LLMs to provide more detailed explanations for the KnowTrace rationales distilled by the backtracking mechanism, thereby stimulating more effective self-improvement finetuning for KnowTrace*.
>
> Overall, we believe that these directions offer valuable avenues for future work to further enhance our framework. We sincerely appreciate your insightful feedback.

---

> ### Author Response · Authors · 2024-11-18
> **Response to Reviewer ifaQ (3/3)**
>
> > **Q4: How many numbers of iterations does KnowTrace adopted in the experimental results of Table 1?**
>
> We understand your concern about the inference overhead of our framework. To effectively address this, we include **a detailed cost and latency analysis** for KnowTrace and two representative baselines (i.e., IRCoT and ERA-CoT). The statistics are summarized as follows:
>
>
> | Dataset | Method                                                 | #Iteration  | #Token (k)  | #Time (s) |
> | :------------ | :------------------------------------------------------------ | :------------: | :------------: | :----------: |
> | HotpotQA | IRCoT  | 3.2 | 1.2 |    5     |
> | | ERA-CoT                      |     1.0    | 2.1  |    13     |
> | | KnowTrace (ours)                                               |     2.5   | 1.4   |  6  |
> | 2Wiki | IRCoT  | 2.8 | 1.5 |    6      |
> | | ERA-CoT                      |     1.0    | 2.3  |    15      |
> | | KnowTrace (ours)                                               |     2.4   | 1.5   |  6  |
> | MuSiQue | IRCoT  | 4.6 | 1.7 |    8      |
> | | ERA-CoT                      |     1.0    | 2.4  |    16      |
> | | KnowTrace (ours)                                               |     3.8   | 1.8   |  9  |
>
> **#Iteration**: Average number of inference iterations per question
> **#Token**: Average number of tokens processed by LLMs per question
> **#Time**: Average inference time per question
>
> We can observe that:
> - Compared to the iterative baseline IRCoT, KnowTrace requires fewer iterations on average, since it can explore **multiple expansion directions** based on the current knowledge structure at each iteration, rather than solely relying on a single chain of thought. This allows KnowTrace to acquire more relevant knowledge in each iteration, reducing the overall number of iterations required.
> - For the restructuring-based baseline ERA-CoT, although it is a non-iterative approach (#Iteration = 1.0), its restructuring process involves 5 LLM-driven steps (entity extraction, relation extraction, relation inference, discrimination, and question answering) for every input question. **These steps are inherently non-parallelizable and all require retrieved passages to be included in the LLM prompts.** Therefore, the resturcturing operations in ERA-CoT incur significantly higher inference time cost than both IRCoT and our KnowTrace.
> - Overall, beyond the iterative and restructuring-based baselines, KnowTrace seamlessly integrates knowledge structuring with multi-step reasoning, enhancing inference performance **without sacrificing the efficiency**. In other words, **KnowTrace achieves a favorable balance of computational cost and multi-step reasoning capability compared to both iterative and restructuring-based baselines.**
>
> At the same time, we would like to respectfully clarify that the backtracing mechanism naturally leverages the knowledge structures organized during KnowTrace inference **without additional LLM calls**. This mechanism produces high-quality rationales for self-improvement training, whose cost aligns with standard parameter-efficient fine-tuning (approximately 2–3 hours on a single NVIDIA A100 GPU).
>
>
>
> **References:**
> [1] Shi et al. _Large Language Models Can Be Easily Distracted by Irrelevant Context_. ICML 2023.
> [2] Wei et al. _InstructRAG: Instructing Retrieval-Augmented Generation via Self-Synthesized Rationales_. 2024.

---

> ### Author Response · Authors · 2024-11-25
> **A Gentle Reminder**
>
> Dear Reviewer ifaQ,
>
> Thank you very much for your time and insightful feedback. We have addressed your questions and comments with detailed clarifications and additional experiments. We would greatly appreciate it if you could reconsider your rating. Additionally, we are more than happy to answer any further questions you may have.
>
> Our responses can be outlined as follows:
> - First, we conducted additional experiments to compare KnowTrace with the **iterative version** of the restructuring-based baseline ERA-CoT, and found that **our framework exhibits the dual advantages in performance and efficiency**.
> - Second, we followed your suggestion to include **a specific example of KnowTrace inference and backtracing mechanism**, which complements the high-level presentation in Figure 1.
> - Third, we discussed **the potential of further integrating our framework with the insights from the other related works**, which offers valuable avenues for future study.
> - Last but not least, we presented **a detailed cost and latency analysis** for our KnowTrace and two representative baselines, aiming to **address your concern about the inference overhead of our framework**.
>
> Best regards,
> Authors

---

> ### Comment · Reviewer_ifaQ · 2024-12-02
> **Response to Author Rebuttals**
>
> Thank the authors for their clear responses and additional experiments! The answers address much of my original concerns, especially relative to the cost/overhead of inference process. I have raised the score accordingly.
>
> I want to thank the authors again for their great effort in responding to my questions.

---

### Official Review · Reviewer_YmpL · 2024-11-06

**Soundness:** 3
**Presentation:** 3
**Contribution:** 3
**Rating:** 6
**Confidence:** 4

**Summary:**

This paper focuses on incorporating structured knowledge (i.e., relational triplets of (subject, relation, object)) in iterative retrieval-augmented generation (RAG) to guide retrievals and facilitate multi-hop reasoning. For each question, a specific knowledge graph (KG) is progressively constructed from the retrieved documents across multiple retrieval iterations until a final answer can be derived. The authors also introduce a reflective mechanism called knowledge backtracking, which identifies the correct trajectories in the KG that lead to correct answers and fine-tunes the LLM to better construct the KG based on these trajectories. Experiments show that the proposed method outperforms selected baseline methods on three multi-hop QA tasks.

**Strengths:**

- Constructing knowledge graphs (KGs) from retrieved documents to guide multi-hop reasoning in RAG is an interesting idea.
- The knowledge backtracking mechanism provides an explicit way to train the LLM to learn intermediate reasoning steps (i.e., KG trajectories) before reaching the final answer.
- Experiments show that the proposed method outperforms baseline methods on three multi-hop QA benchmarks (although evaluated on a small subset of test samples).

**Weaknesses:**

- Potentially limited practical value. My major concern is about the inference efficiency of the RAG system, as the proposed method requires constructing a knowledge graph (KG) for each question through iterative retrievals, which seems time-consuming. For example, with K = 5 documents retrieved per iteration, how many iterations are needed for each question? And what is the average inference time per question? A latency study comparing the proposed method with baseline methods is needed to validate its practical applicability.

- Unclear generalizability. As described in Section 4.1, only 500 questions are randomly sampled as the test set in each benchmark, which represents only a small portion of the entire test set (e.g., 2WikiMultiHop has 12,576 test samples in total), and thus the evaluation results may not be representative. Would the findings still hold if the evaluation were conducted on the entire test set? Furthermore, does the proposed method generalize to other open-domain QA tasks (e.g., NaturalQuestions/TriviaQA), which do not heavily rely on multi-hop reasoning? Would it still outperform baselines in such cases? Given the current simple evaluation setting, the generalizability of the proposed method remains unclear.

**Questions:**

Please address the technical questions in weaknesses.

Below are some clarification questions on self-taught fine-tuning:
1. In Figure 1, how are rationales generated given the correct trajectories (i.e., connected knowledge triplets)? Does the process involve prompting the LLM to generate a rationale that explains how these triplets lead to the final answer?
2. In Algorithm 2, what is the input/output data format in $\mathcal{D}_{z}$ used to fine-tune the model?

---

> ### Author Response · Authors · 2024-11-18
> **Response to Reviewer YmpL (1/3)**
>
> We sincerely appreciate your careful reading and valuable comments on our paper. Below, we provide a detailed response to your concerns and questions.
>
> > **Q1: Inference efficiency of KnowTrace (Cost and Latency Study).**
>
> We understand your concern about the inference overhead of our framework. To effectively address this, we include **a detailed cost and latency analysis** for KnowTrace and two representative baselines (i.e., IRCoT and ERA-CoT). The statistics are summarized as follows:
>
> | Dataset | Method                                                 | #Iteration  | #Token (k)  | #Time (s) |
> | :------------ | :------------------------------------------------------------ | :------------: | :------------: | :----------: |
> | HotpotQA | IRCoT  | 3.2 | 1.2 |    5     |
> | | ERA-CoT                      |     1.0    | 2.1  |    13     |
> | | KnowTrace (ours)                                               |     2.5   | 1.4   |  6  |
> | 2Wiki | IRCoT  | 2.8 | 1.5 |    6      |
> | | ERA-CoT                      |     1.0    | 2.3  |    15      |
> | | KnowTrace (ours)                                               |     2.4   | 1.5   |  6  |
> | MuSiQue | IRCoT  | 4.6 | 1.7 |    8      |
> | | ERA-CoT                      |     1.0    | 2.4  |    16      |
> | | KnowTrace (ours)                                               |     3.8   | 1.8   |  9  |
>
> **#Iteration**: Average number of inference iterations per question
> **#Token**: Average number of tokens processed by LLMs per question
> **#Time**: Average inference time per question
>
> We can observe that:
> - Compared to the iterative baseline IRCoT, KnowTrace requires fewer iterations on average, since it can explore **multiple expansion directions** based on the current knowledge structure at each iteration, rather than solely relying on a single chain of thought. This allows KnowTrace to acquire more relevant knowledge in each iteration, reducing the overall number of iterations required.
> - For the restructuring-based baseline ERA-CoT, although it is a non-iterative approach (#Iteration = 1.0), its restructuring process involves 5 LLM-driven steps (entity extraction, relation extraction, relation inference, discrimination, and question answering) for every input question. **These steps are inherently non-parallelizable and all require retrieved passages to be included in the LLM prompts.** Therefore, the resturcturing operations in ERA-CoT incur significantly higher inference time cost than both IRCoT and our KnowTrace.
> - Overall, beyond the iterative and restructuring-based baselines, KnowTrace seamlessly integrates knowledge structuring with multi-step reasoning, enhancing inference performance **without sacrificing the efficiency**. In other words, **KnowTrace achieves a favorable balance of computational cost and multi-step reasoning capability compared to both iterative and restructuring-based baselines.**
>
> At the same time, we would like to respectfully clarify that the backtracing mechanism naturally leverages the knowledge structures organized during KnowTrace inference **without additional LLM calls**. This mechanism produces high-quality rationales for self-improvement training, whose cost aligns with standard parameter-efficient fine-tuning (approximately 2–3 hours on a single NVIDIA A100 GPU).

---

> ### Author Response · Authors · 2024-11-18
> **Response to Reviewer YmpL (2/3)**
>
> > **Q2: Generalizability of KnowTrace on entire test set and other open-domain QA tasks.**
>
> We would first like to respectfully clarify that our experiments adopt the same evaluation settings as prior works (e.g., IRCoT, Iter-RetGen, and SG-Prompt) for consistency.
>
> At the same time, we totally understand your concerns regarding the generalizability of our framework. To address this, we follow your suggestion to evaluate KnowTrace (and two representative baselines) on the entire test set of three multi-hop QA datasets as well as two additional open-domain QA datasets. The EM results are summarized as follows:
>
> | Method                                                 | HotpotQA (7,405 questions)  | 2Wiki (12,576 questions) | MuSiQue (2,417 questions) | NQ (3,610 questions) | TQA (11,313 questions) |
> | :------------------------------------------------------------ | :------------: | :------------: | :----------:  | :----------:  | :----------: |
> | IRCoT  | 0.382 | 0.352 |  0.207   | 0.554 | 0.706 |
> | ERA-CoT           | 0.405   | 0.368  |  0.224  | 0.568 | 0.729 |
> | KnowTrace (ours)  | 0.443   | 0.394 | 0.265  | 0.573 | 0.725 |
>
> We can observe that:
> - On the full test sets of the three standard multi-hop QA datasets, KnowTrace maintains superior performance, thereby confirming the effectiveness and generalizability of our framework on the multi-hop QA task.
> - While our work is primarily designed for the complex multi-hop QA task, KnowTrace still outperforms IRCoT and performs comparably to ERA-CoT on the two simpler open-domain QA datasets, i.e., NaturalQuestions (NQ) and TriviaQA (TQA).
> - These results comprehensively confirm that **our framework can effectively handle complex multi-hop questions without compromising performance on simpler ones**.
>
> > **Q3: How are the rationales generated given the correct trajectories in Figure 1? Does this process involve a new LLM invocation?**
>
> For the backtracing mechanism, the reasoning rationales are distilled by tracing back along the self-organized knowledge structures from the target entities to the initial entities as described in Section 3.3. In other words, this process is inherently built upon the knowledge structures acquired during KnowTrace inference, and **does not require any additional LLM invocations**.
>
> To address your concern more clearly, we provide **a detailed example of KnowTrace inference and backtracing** below.
>
> **Example of KnowTrace Inference:**
> ```
> Input Question: Where was the person who wrote about the rioting being a dividing factor in Birmingham educated?
>
> [Iteration 1]
> ### Knowledge Memory: None
> ### Knowledge Exploration
> Whether the collected knowledge triplets are sufficient: No
> What entity-relation pairs to retrieve in the current step:
> - The rioting being a dividing factor in Birmingham: Find out who wrote about this topic.
> - Birmingham: Find out where Birmingham is located.
> ### Retrieve Textual Passages From Wikipedia
> ### Knowledge Completion
> (James Watt, wrote about, the rioting being a dividing factor in Birmingham);
> (James Watt, is, an industrialist);
> (Birmingham, is located in, the West Midlands region of England)
>
> [Iteration 2]
> ### Knowledge Memory: The Knowledge Triplets Acquired in Iteration 1
> ### Knowledge Exploration
> Whether the collected knowledge triplets are sufficient: No
> What entity-relation pairs to retrieve in the current step:
> - James Watt: Find out which school James Watt attended.
> ### Retrieve Textual Passages From Wikipedia
> ### Knowledge Completion
> (James Watt, was educated at, University of Glasgow)
>
> [Iteration 3]
> ### Knowledge Memory: The Knowledge Triplets Acquired in Iterations 1 and 2
> ### Knowledge Exploration
> Whether the collected knowledge triplets are sufficient: Yes
> Thought: James Watt wrote about the rioting being a dividing factor in Birmingham. He was educated at the University of Glasgow.
> Answer: University of Glasgow
> ```
>
> **Example of Backtracing Mechanism:**
> Based on the transparent knowledge structure traced by KnowTrace in the above example, we can naturally backtrace from the answer entity _University of Glasgow_ to identify the following evidence subgraph: **(James Watt, wrote about, the rioting being a dividing factor in Birmingham); (James Watt, was educated at, University of Glasgow)**. In this way, our framework naturally allows for filtering unavailing exploration (e.g., "- Birmingham: Find out where Birmingham is located") and extraneous completion (e.g., (James Watt, is, an industrialist)) from the LLM generations, thereby presenting higher-quality reasoning rationales for the self-improvement finetuning.

---

> > ### Author Response · Authors · 2024-11-18
> > **Response to Reviewer YmpL (3/3)**
> >
> > > **Q4: What is the input/output data format in $\mathcal{D}_z$ used to finetune the model in Algorithm 2?**
> >
> > The self-improvement finetuning dataset $\mathcal{D}_z$ consists of the input-output pairs distilled from the reasoning traces of KnowTrace using the backtracing mechanism (i.e., filtering out the unavailing exploration and extraneous completion for higher-quality supervision).
> >
> > Specifically, the distilled input-output format in $\mathcal{D}_z$ for the knowledge exploration phase is:
> > ```
> > - Input: The same exploration prompt used during inference.
> > - Output: Entity-relation guidances that lead to the supportive triplets, such as "- The rioting being a dividing factor in Birmingham: Find out who wrote about this topic" in the example provided above.
> > ```
> >
> > For the knowledge completion phase, the distilled input-output format in $\mathcal{D}_z$ is:
> > ```
> > - Input: The same completion prompt used during inference.
> > - Output: knowledge triplets that support the inference of the final answer, such as "(James Watt, wrote about, the rioting being a dividing factor in Birmingham)" in the example provided above.
> > ```
> >
> > We sincerely appreciate your insightful comments, which, we believe, are invaluable for improving our paper. We hope our responses can adequately address your concerns. If you have any further questions or suggestions, we would be more than happy to discuss them with you.

---

> > > ### Comment · Reviewer_YmpL · 2024-11-25
> > >
> > > Thank the authors for answering my questions. I’ve also read the other reviewers’ comments, which didn’t change my opinion on this paper. Therefore, I will maintain my score.

---

### Author Response · Authors · 2024-11-22
**Looking Forward to Your Feedback**

Dear reviewers,

We sincerely thank all reviewers for their careful reading and valuable comments on our paper.

Since **the discussion deadline is approaching**, we look forward to hearing your feedback on our responses.

We would appreciate the chance to address any remaining concerns that you may still have.

Best regards,
Submission654 Authors

---

### Author Response · Authors · 2024-11-26
**General Response**

We sincerely thank all reviewers for their careful reading and valuable comments on our paper. We are pleased that all reviewers have expressed a positive stance towards our work, and we greatly appreciate the opportunity to address the reviewers' concerns and further improve our manuscript.

In this post, we would like to summarize the identified strengths of our work, outline the revisions we have made in the updated version, and highlight the insights we expect to provide.

---

Strengths:

- **Motivation**: "_The paper is well-motivated_" (Reviewer ```ifaQ```). "_The method motivation is clear, which is restructuring the retrieved passages to facilitate better reasoning._" (Reviewer ```8nP3```).
- **Method Novelty**: "_Constructing knowledge graphs (KGs) from retrieved documents to guide multi-hop reasoning in RAG is an interesting idea_" (Reviewer ```YmpL```). "_This paper introduces a clever incorporation of knowledge graph operation for structured RAG_" (Reviewer ```ifaQ```). "_The proposed method is straightforward, intuitive, and easy to implement_"; "_It is innovative that the paper leverages the structured nature of reasoning paths to filter and refine generated trajectories for model training_" (Reviewer ```Tjwb```).
- **Performance**: "_The paper targets a challenging multi-hop QA setting in RAG and presents a new framework supported by strong empirical performance_"; "_The paper includes multiple, comprehensive baselines to support the main experimental results of the proposed approach_" (Reviewer ```ifaQ```). "_Experiments show that the proposed method outperforms baseline methods_" (Reviewer ```YmpL```). "_The method demonstrates strong empirical performance across multiple datasets compared to the baseline methods._" (Reviewer ```Tjwb```). "_KnowTrace demonstrates stable improvement across all datasets and setups_"; "_The self-improvement loop shows promising results on scaling such RAG methods_" (Reviewer ```8nP3```).
- **Writing Quality**: "_The paper is generally well-written_" (Reviewer ```ifaQ```). "_The paper is well-written, clear, and easy to follow_" (Reviewer ```Tjwb```).

---

In the updated manuscript, we:

- follow the suggestions of Reviewers ```YmpL``` and ```8nP3``` to present a detailed cost and latency analysis, showing that **our framework exhibits the dual advantages in performance and efficiency**.
- include a specific example of KnowTrace inference and backtracing mechanism, which **complements the high-level presentation** in Figure 1 (Reviewers ```ifaQ```).
- evaluate KnowTrace on the entire test set of three multi-hop QA datasets as well as two additional open-domain QA datasets, **demonstrating the generalizability of our work** (Reviewers ```YmpL```).
- provide a detailed discussion on the key distinctions between our framework and the prior works, **highlighting the unique contributions of our work** (Reviewer ```Tjwb```).

---

In summary, our work is expected to contribute two new insights to the research community on retrieval-augmented generation:

- **Explicit Knowledge Tracing for Inference**: From the perspective of explicit knowledge tracing, we can seamlessly organize a transparent knowledge structure for each input question throughout the multi-step reasoning process, _endowing the LLM with an intelligible context to facilitate its inference capability (without incurring significant overhead)._
- **Natural Knowledge Backtracing for Post-Training**: The transparent knowledge structures (traced out during inference) naturally allow us to backtrace high-quality reasoning rationales from positive trajectories, _which can be leveraged for effective post-training in a self-taught manner._

---

---

### Meta-Review · Area_Chair_dog7 · 2024-12-13

**Metareview:**

This paper proposed a paradigm, KnowTrace, for explicit knowledge tracing by acquiring knowledge triplets through iterative retrievals and tracing out a knowledge graph. The reviewers generally found the proposed method to be interesting and intuitive but also raised concerns regarding (1) the practicality and complexity of the method (reviewers found the method to be too complicated and may generalize poorly), and (2) its novelty compared to existing works. Specifically, the reviewers believed that KnowTrace bears similarities with existing RAG approaches that integrate reason processes like [1] and [2] (I think both works are quite related to this work). The authors didn't incorporate these related works into their updated paper, while I agree with the reviewers that such discussions are necessary to clarify the novelty and contribution of this work and strengthen the paper.

[1] InstructRAG: Instructing Retrieval-Augmented Generation via Self-Synthesized Rationales
[2] Graph-Guided Reasoning for Multi-Hop Question Answering in Large Language Models

**Additional Comments On Reviewer Discussion:**

This is a borderline paper and the reviewers had a discussion. After the discussion, no reviewer was willing to champion the paper for acceptance and believed that the paper in its current form still has room for improvement, especially because the technical contributions, compared to existing related works, appeared quite marginal.

---

### Decision · Program_Chairs · 2025-01-22

Reject